# Nonergodicity and Simpson's paradox in neurocognitive dynamics of cognitive control

Percy K. Mistry [1,4] ✉, Nicholas K. Branigan [1,4] ✉, Zhiyao Gao [1], Weidong Cai [1,2] & Vinod Menon [1,2,3] ✉

Nonergodicity and Simpson's paradox present significant, yet under-appreciated challenges in cognitive neuroscience. Leveraging brain imaging and behavioral data from over 4000 individuals and a Bayesian computational model of cognitive dynamics, we investigated brain-behavior relationships underlying cognitive control at both between-subjects and within-subjects levels. Strikingly, brain-behavior associations reversed across levels of analysis, revealing pervasive nonergodicity. Within-subjects analysis uncovered dissociated neural representations of reactive and proactive control and revealed that individuals who adaptively versus maladaptively regulated cognitive control exhibited distinct brain-behavior associations. Our findings demonstrate that between-subjects analyses can fundamentally mischaracterize within-individuals mechanisms, as group-level patterns not only disagreed with individual-level patterns but often reversed them. This work highlights the necessity of distinguishing between-subjects and within-subjects inferences in neuroscience, with implications for understanding cognitive mechanisms and designing personalized interventions.

Ergodicity—the equivalence between time and ensemble averages—has been a cornerstone in physics since the nineteenth century, shaping how we connect micro-dynamics to macro regularities[1–3]. In recent years, a parallel question has emerged in psychology[4,5], economics[6], and neuroscience[7]: do group-level averages accurately represent individual-level processes? In psychology and neuroscience, an ergodic process is one for which patterns observed across individuals (from between-individuals variation) are equivalent to patterns observed within individuals over time (from within-individuals variation)[8]. Nonergodicity occurs when this equivalence breaks down: analyses at the between-individuals level cannot be used to make inferences about processes at the within-individuals level, and vice versa[4,9].

An important example of nonergodicity is Simpson's paradox[10,11], exemplified in psychology by the speed-accuracy tradeoff[12]: speed and accuracy are often positively correlated between individuals (faster people are more accurate)[13], but are negatively correlated within individuals (when an individual tries to respond faster, their accuracy decreases)[14]. This striking divergence between group-level and individual-level patterns illustrates that between- and within-individuals analyses answer fundamentally different scientific questions[8]. This example also shows that both between- and within-individuals questions can be of scientific interest. However, many important questions in the study of the brain and behavior are posed at the within-individuals level[15], and answering these questions requires either within-individuals analyses or the strong assumption of ergodicity[4,5].

For a process to be ergodic, it must satisfy two key conditions: (i) homogeneity, each individual follows the same statistical model, and (ii) stationarity, the process's statistical properties are invariant across time[9]. When either condition fails, patterns observed across

[1]Department of Psychiatry & Behavioral Sciences, Stanford University School of Medicine, Stanford, CA, USA. [2]Wu Tsai Neurosciences Institute, Stanford University, Stanford, CA, USA. [3]Department of Neurology & Neurological Sciences, Stanford University School of Medicine, Stanford, CA, USA. [4]These authors contributed equally: Percy K. Mistry, Nicholas K. Branigan. ✉e-mail: percym@stanford.edu; branigan@stanford.edu; menon@stanford.edu

individuals need not reflect, and can even contradict, mechanisms operating within individuals[4,7,9]. Given the ubiquity of individual differences (heterogeneity) and temporal instability in behavior (nonstationarity), theoretical arguments suggest that cognitive processes may be inherently nonergodic[8].

The implications of nonergodicity in neural and behavioral science are profound. Analyzing data from a single individual over time can yield insights that cannot be obtained by analyzing data across multiple individuals at a single point in time[4,5,16]. This challenges traditional assumptions about the generalizability of findings from between-individuals analyses to individual processes[4,8,12,16–18], fundamentally changing how we interpret results. While progress has been made in understanding nonergodicity in behavior[4,5,9,15,19,20] and establishing strong theoretical frameworks[7,17,21,22], few studies have empirically addressed nonergodicity in human brain function[7,23,24].

This gap in research is particularly striking in human cognitive neuroscience, where most studies of brain-behavior relations rely heavily on between-subjects analysis. In the presence of nonergodicity, the association between neural activity and behavior may differ substantially when examined between subjects after averaging over time (Fig. 1a) compared to within subjects across time (Fig. 1b), potentially leading to a neurocognitive manifestation of Simpson's paradox (Fig. 1c)[9,12,17]. This phenomenon has implications for our understanding of brain function and cognition. Group-level relationships may not provide information about the dynamic processes occurring at the individual level, and can lead to incomplete or even misleading conclusions about the neural mechanisms underlying cognitive processes[4,5,9,15,19,20].

To address critical gaps in our knowledge of nonergodicity in neurocognitive processes, we leveraged a large sample ($N \sim 4000$) from the Adolescent Brain Cognitive Development (ABCD) study, a multisite study of brain development and behavior in children and adolescents[25]. We used behavioral and brain imaging data to examine the neural mechanisms of cognitive control at both the between-subjects and within-subjects levels, aiming to elucidate the ergodicity of these processes and implications for our understanding of brain function and cognition.

An important component of human cognition is inhibitory control, the ability to withhold or cancel maladaptive actions, thoughts, and emotions. This cognitive control function is a fundamental component of goal-directed behavior[26,27], allowing individuals to navigate complex environments, adapt to changing circumstances, and maintain focus on long-term goals in the face of immediate temptations or distractions[28]. Given its central role in cognitive control, understanding the neural mechanisms underlying inhibitory control has been a major focus of cognitive neuroscience research[29]. Inhibitory control engages a distributed network of cortical and subcortical regions. Previous studies have implicated the involvement of the salience network, particularly the anterior insula and dorsal anterior cingulate cortex, in detecting and processing relevant stimuli and coordinating neural resources for inhibitory control[30–34]. Additionally, the inferior frontal gyrus, presupplementary motor area, and basal ganglia have been shown to play crucial roles in implementing response inhibition[35–41]. While the dynamic interplay between these cortical and subcortical regions is known to underpin successful inhibitory control, our understanding of how these neural processes relate to moment-to-moment fluctuations in cognitive control at the individual level remains limited.

The stop signal task (SST) is a widely used paradigm to assess inhibitory control. Participants are instructed to respond quickly to a primary "Go" stimulus but to withhold their response when an infrequent "stop" signal appears[27]. The ability to inhibit responses is quantified using the stop signal reaction time (SSRT), which estimates the latency of the stopping process[27,28]. Conventionally, studies investigating the neural basis of inhibitory control have relied on

between-subjects analyses of SSRT, correlating individual differences in SSRT with individual differences in task-related brain activation[31,41–47]. These between-subjects associations are often implicitly interpreted as reflecting within-subjects mechanisms. However, this implicit generalization from between-subjects to within-subjects levels critically depends on the assumption of ergodicity[5,9,17,48].

A major barrier to understanding the dynamics of cognitive mechanisms has been the difficulty in measuring within-individuals processes. In the study of inhibitory control, for example, traditional models of the SST do not provide trial-wise SSRT estimates, instead requiring a substantial number of trials for a single estimate[49]. Without trial-by-trial measures of cognitive processes, researchers cannot examine within-subjects dynamics directly, making it difficult to generate insights about individual-level processes.

To address these limitations and capture the dynamic nature of inhibitory control, we recently developed the Proactive Reactive and Attentional Dynamics (PRAD) model of SST behavior, incorporating proactive and reactive control mechanisms[50]. The PRAD model allowed us to infer dynamic, trial-level measures corresponding to elemental neurocognitive subprocesses for each subject, including SSRTs and measures of proactive delaying based on stop signal anticipation. PRAD parameters characterize the latent cognitive constructs governing action execution and inhibition in the SST, providing a more comprehensive view of inhibitory control processes than traditional models.

The PRAD model dissociates observed behavior into elemental latent neurocognitive subprocesses that can characterize multiple dynamic measures of within-subjects variation. This is critical to conducting interpretable within-subjects analysis, since behavioral variability within a cognitive task is often governed by the complex interplay of multiple neurocognitive subprocesses, each of which may potentially have distinct underlying brain dynamics. By considering the various sequential dependencies and compensatory, competing, and interacting neurocognitive mechanisms underlying cognitive control, the PRAD model dissociates elemental processes and improves the interpretability of inferred within-subjects brain-behavior relationships. Specifically, the PRAD model can infer non-random and sequentially dependent trial-level variability in the probability and duration of proactive delayed responding[51], as well as in the effect of attentional modulation on the reactive SSRT.

We combined task fMRI data with the PRAD model to investigate the relationship between task-evoked brain responses and dynamic inhibitory control processes at both between-subjects and within-subjects levels. This approach allows us to directly compare inferences made from traditional group-level analyses (group-level associations) with those derived from a fine-grained examination of within-individuals variation in neurocognitive dynamics. We leveraged the dynamic representations of subjects' behaviors from the PRAD cognitive model alongside simultaneous task fMRI data to probe how moment-to-moment variations in brain activity relate to fluctuations in inhibitory control processes. This analysis strategy promotes a richer understanding of the neural basis of inhibitory control, potentially revealing insights unavailable from conventional group-level analyses.

We pursued four interconnected aims (Fig. 1d). First, we examined how observed overt behavioral measures and latent PRAD model parameters[50] related to brain activity at between-subjects and within-subjects levels. By comparing these levels, we demonstrate the limitations of generalizing inferences about inhibitory control mechanisms from between-subjects to within-subjects analysis. Second, we leveraged the within-subjects approach to probe how the brain implements proactive and reactive control processes underlying inhibitory control. Proactive control involves anticipation and preparation for stopping, while reactive control involves actual implementation of response inhibition[52]. A key theoretical question is whether these two forms of control rely on shared or distinct neural

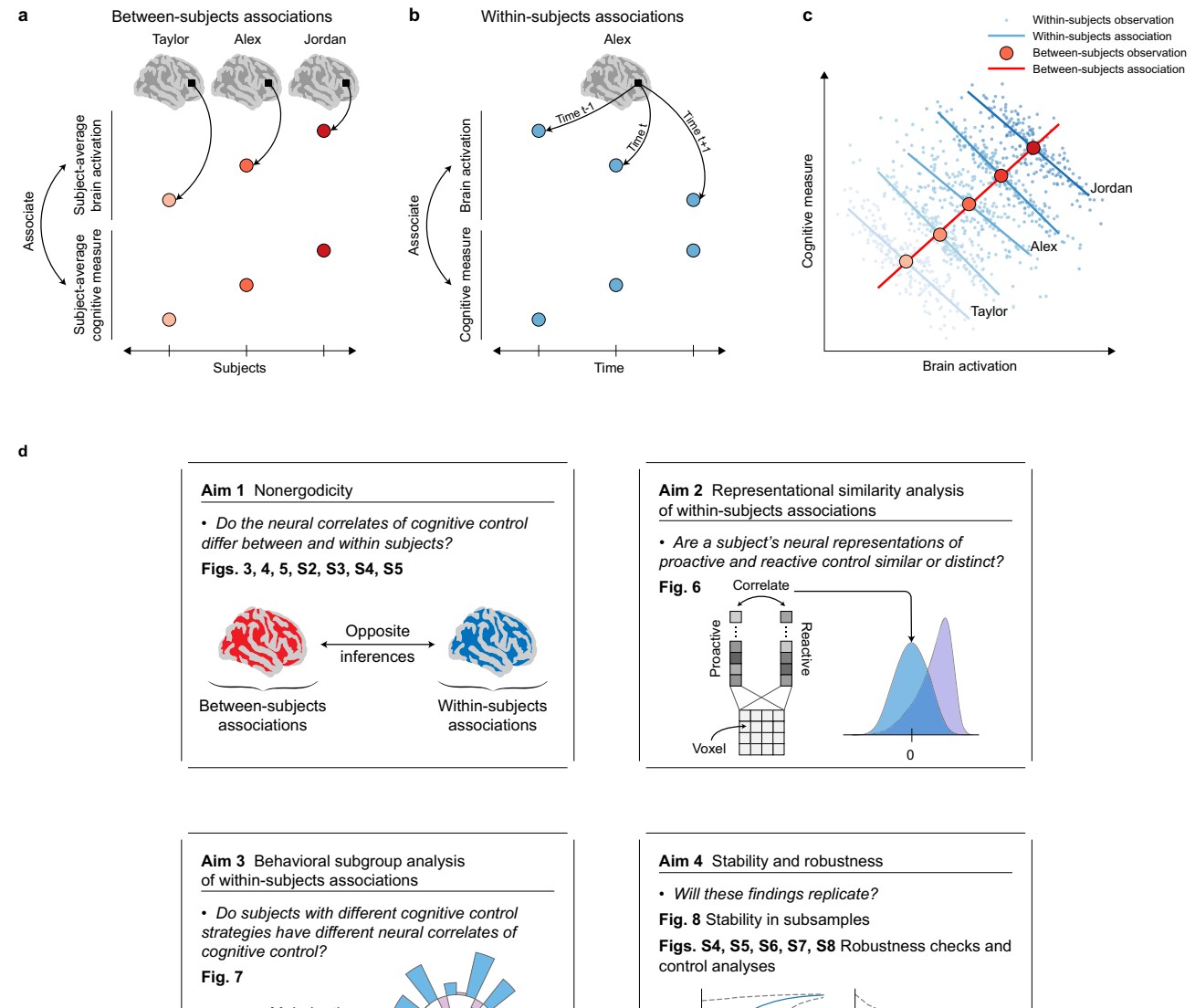

**Fig. 1 | Conceptual overview of the study and key findings.** The figure illustrates the methodology for between-subjects and within-subjects analyses, the concept of nonergodicity, and the study's aims. **a** Between-subjects analysis. Subject-average brain activation in each voxel is correlated with a subject-average cognitive measure across the population. **b** Within-subjects analysis. For each individual, the time-series of brain activity in each voxel is associated with the time-series of a cognitive measure. **c** Simpson's paradox. Simpson's paradox occurs when associations disagree between subjects and within subjects; it exemplifies nonergodicity in the brain and behavioral sciences. **d** Study aims. We examined brain-behavior substrates. By examining the similarity between within-subjects brain representations of proactive and reactive processes across different brain networks, we tested whether these control mechanisms are neurally dissociated or overlapping. This analysis advances understanding of the neural architecture underlying inhibitory control and clarifies the relationship between these two control mechanisms in the context of nonergodicity. Third, we identified subgroups based on individual differences in adaptive regulation of inhibitory control, examining whether within-subjects brain-behavior associations differ associations for nonergodicity; used within-subjects associations to probe the brain implementations of proactive and reactive control as well as adaptive cognitive control strategies; and tested our within-subjects results for stability and robustness. Nonergodic patterns in brain-behavior associations were consistently observed, revealing that group-level (between-subjects) and individual-level (within-subjects) associations yield divergent results for cognitive control processes. This challenges the common assumption that findings from such group-level analyses can be directly applied to understand individual-level cognitive processes.

across these subgroups. This approach allowed us to investigate whether individuals' brain-behavior associations vary depending on their cognitive strategies. Finally, motivated by the replicability crisis in neuroscience research, we extensively assessed the stability and robustness of our findings.

Our findings reveal pervasive nonergodicity in the neurocognitive dynamics of inhibitory control, demonstrating a stark contrast between traditional between-subjects analyses and novel within-subjects analyses that account for dynamic cognitive processes.

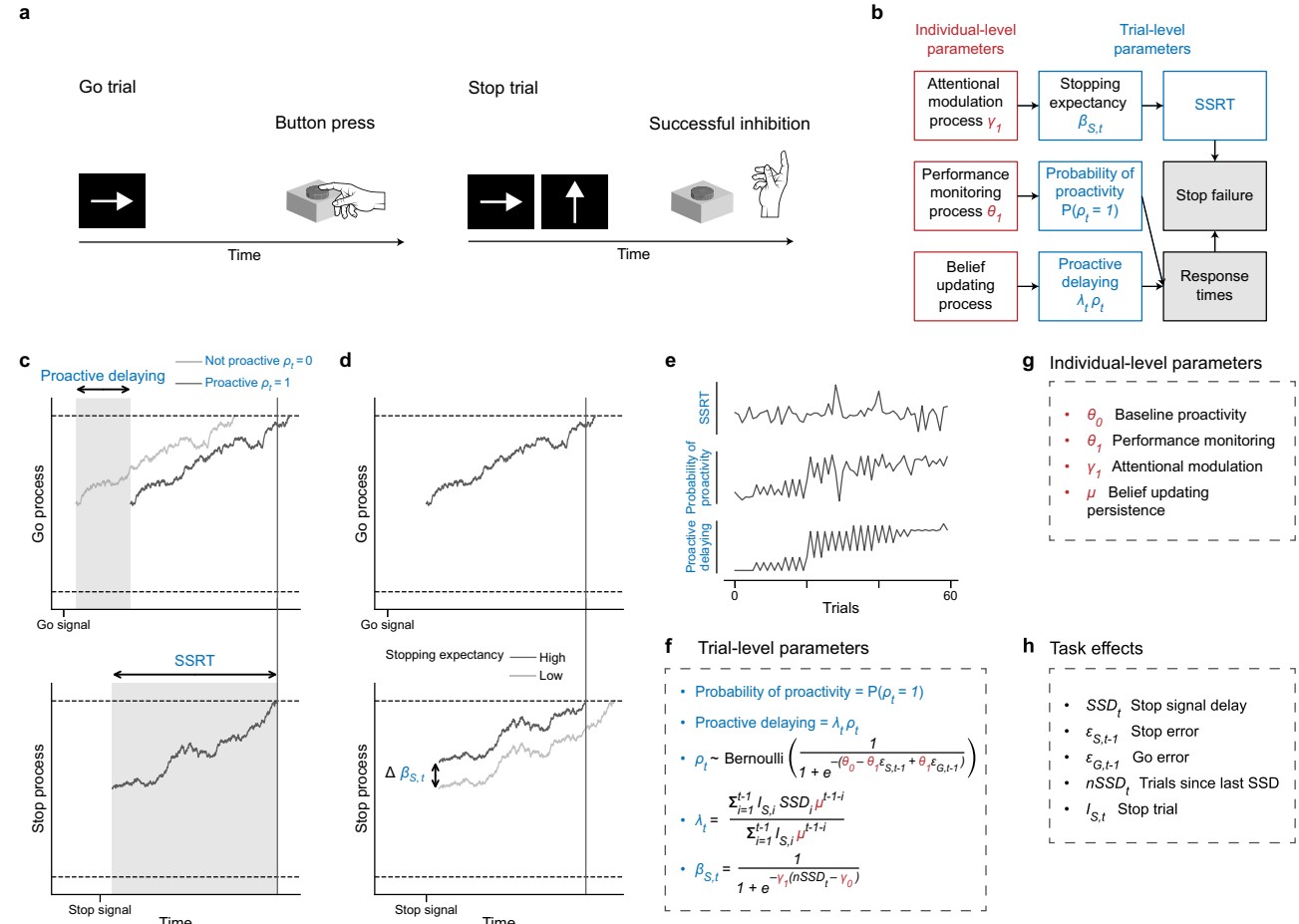

**Fig. 2 | Computational modeling of inhibitory control dynamics in the stop signal task. a** Stop signal task. On go trials subjects should respond by pressing a button to indicate the arrow direction, and on stop trials should inhibit their response when the stop signal appears. **b** Computational model schematic. The computational model infers individual-level and trial-level parameters that measure inhibitory control from observed stop failures and response times. SSRT is the time it takes for the stop process to complete. Probability of proactivity is the probability that a subject uses a proactive control strategy. Proactive delaying is the delay in initiating the go process when a subject uses a proactive control strategy.

**c**, **d** Within-trial dynamics. **c** Increased proactivity improves stop trial performance through its effect on the go process. **d** Increased stopping expectancy improves stop trial performance through its effect on SSRT. **e** Trial-by-trial dynamics. Illustration of how SSRT, probability of proactivity, and proactive delaying might vary over time within an individual. **f**–**h** Key model quantities and their relations. The computational model allows for a detailed, dynamic analysis of inhibitory control processes with trial-level temporal resolution, enabling the investigation of within-subjects variability and nonergodic patterns in cognitive control.

Results not only highlight the limitations of assuming ergodicity in cognitive control research, but also provide insights into the neural mechanisms underlying inhibitory control at the individual level. Specifically, our within-subjects analysis reveals how proactive and reactive control processes are implemented in the brain through time, how these processes are neurally dissociated, and how individual differences in cognitive strategies shape brain-behavior relationships—insights entirely unavailable from conventional between-subjects approaches. By demonstrating substantive differences between group-level and individual-level neurocognitive dynamics, our work underscores the necessity of embracing nonergodicity in cognitive neuroscience.

## Results

### Dynamic cognitive process model of behavior
We used data from the SST (Fig. 2a) to investigate dynamic cognitive processes underlying cognitive control in a large sample ($N\approx4000$) from the baseline visit of the ABCD study. We developed PRAD, a dynamic cognitive process model[50] that provides a multidimensional perspective of the elemental cognitive processes governing the

reactive and proactive dynamics involved in inhibitory control. This model incorporates latent dynamics that respond to internal cognitive states (endogenous variables) and external environmental contingencies (exogenous variables), with interaction between these dynamics governed by latent trait measures. The latent trait (individual-level) and dynamic (trial-level) measures are simultaneously inferred within a hierarchical Bayesian framework (Fig. 2b). For each subject and trial, the model infers a measure of reactive inhibitory control, SSRT, and 2 measures of proactive control, probability of proactivity and proactive delaying (Fig. 2c, d). SSRT measures how long it takes for a subject to inhibit a response, probability of proactivity measures whether they use a strategy of proactive delaying, and proactive delaying measures the length of proactive delays when a proactive strategy is utilized[51,53]. Importantly, all 3 of these parameters are inferred at the trial level, with SSRT inferred for each stop trial and probability of proactivity and proactive delaying inferred for each go and stop trial (Fig. 2e). The SST consisted of 360 trials, of which 60 were stop trials and 300 were go trials. SI Fig. S1 provides a detailed illustration of the model. Details of the PRAD model are in the "Methods", and further validation can be found in a separate study[50].

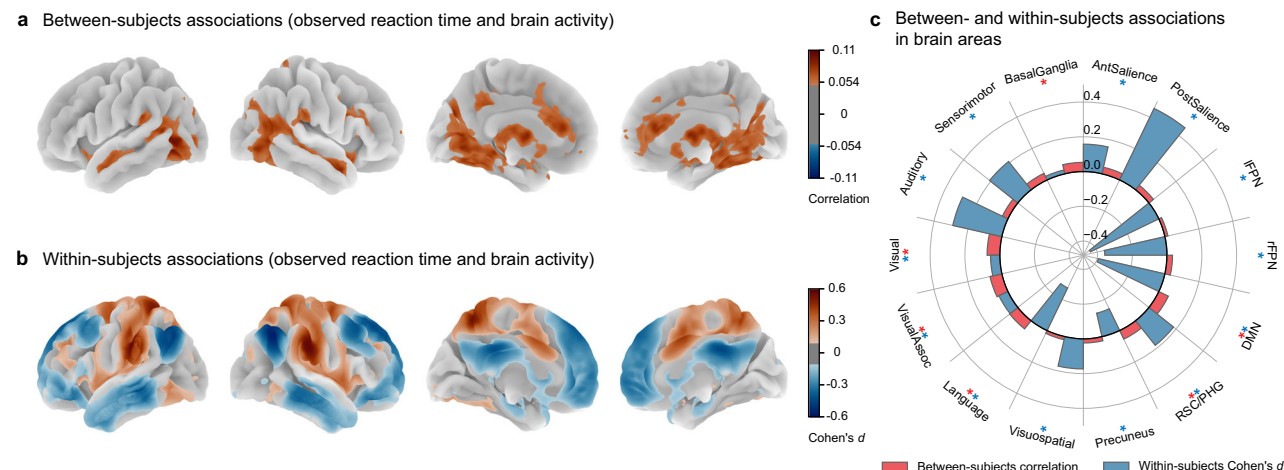

**Fig. 3 | Nonergodicity is exhibited by a directly observed behavioral measure.** **a** Between-subjects analysis. Whole-brain correlation map of associations between subject-average go reaction time and brain activation (correct stop versus correct go activation). Thresholded at Pearson $r \geq 0.05$. **b** Within-subjects analysis. Whole-brain Cohen's $d$ map of associations between trial-by-trial go reaction time and brain activity. Thresholded at Cohen's $d \geq 0.1$. **c** Network-level comparison. Between- and within-subjects associations of go reaction time and brain activity were visualized in brain areas. Statistical significance is indicated by colored asterisks: red for between subjects ($P_{FDR} < 0.01$) and blue for within subjects ($P_{FDR} < 0.01$). For both between- and within-subjects analyses: $N = 4423$. Go reaction time showed divergent between- and within-subjects associations. The widespread differences, and even sign reversals, between levels of analysis provide evidence that brain-behavior relationships underlying inhibitory control are nonergodic. AntSalience anterior salience network, PostSalience posterior salience network,

lFPN left frontoparietal network, rFPN right frontoparietal network, DMN default mode network, RSC/PHG retrosplenial cortex/parahippocampal gyrus network, Precuneus precuneus network, Visuospatial visuospatial network, Language language network, VisualAssoc visual association network, Visual visual network, Auditory auditory network, Sensorimotor sensorimotor network, BasalGanglia basal ganglia network. Two-sided permutation tests were used with FDR correction. Between-subjects $P_{FDR}$: AntSalience = 0.018, PostSalience = 0.028, lFPN = 0.26, rFPN = 0.041, DMN = 0.0014, RSC/PHG = 0.0014, Precuneus = 0.14, Visuospatial = 0.26, Language=0.0017, VisualAssoc = 0.0014, Visual = 0.0014, Auditory = 0.025, Sensorimotor = 0.010, BasalGanglia = 0.0037. Within-subjects $P_{FDR}$: AntSalience < 0.001, PostSalience < 0.001, lFPN < 0.001, rFPN < 0.001, DMN < 0.001, RSC/PHG < 0.001, Precuneus < 0.001, Visuospatial < 0.001, Language < 0.001, VisualAssoc < 0.001, Visual < 0.001, Auditory < 0.001, Sensorimotor < 0.001, BasalGanglia = 0.16.

## Between-subjects and within-subjects analysis of brain-behavior associations

We examined how 1 behavioral measure that was directly observed at a trial level—go trial reaction time (RT) (Fig. 3)—and 3 key parameters of the PRAD model that were inferred at a trial level—SSRT, probability of proactivity, and proactive delaying (Figs. 4 and 5)—related to brain activity at both between-subjects and within-subjects levels. We conducted both between-subjects and within-subjects analyses using fMRI data from the SST (go RT, SSRT, and probability of proactivity $N = 4423$; proactive delaying $N = 4137$). For the between-subjects analyses, we Pearson correlated subject-average brain activation during successful stopping (correct stop versus correct go activation) with subject-average go RT, SSRT, probability of proactivity, and proactive delaying across participants[31,41,44–47]. For the within-subjects analyses, we regressed the fMRI signal on go RT, SSRT, probability of proactivity, and proactive delaying using fMRI general linear models. We modeled the fMRI signal as a combination of static trial-type effects and dynamic effects proportional to trial-by-trial variations in the cognitive measures. This allowed us to examine how fluctuations in cognitive processes covaried with brain activity within each individual, while adjusting for stimulus types. These complementary approaches enabled us to compare group-level and individual-level brain-behavior relationships.

## Nonergodic brain-behavior associations using observed reaction time

To evaluate differences in brain-behavior relations measured at within-subjects and between-subjects levels with a directly observed metric, we related go RT to brain activity at both levels of analysis. Between subjects, brain-RT associations were almost entirely positive, with effects in salience network regions (bilateral anterior insula, dorsal anterior cingulate cortex), bilateral orbitofrontal cortex, right middle frontal gyrus, medial occipital cortex, basal ganglia, and default mode

network regions (precuneus, posterior cingulate cortex, bilateral middle and inferior temporal gyrus, medial temporal lobe) (Fig. 3a). In contrast, within subjects, RT fluctuations exhibited both prominent positive and negative effects (Fig. 3b and SI Fig. S2a). Extensive positive associations were observed in salience hubs (anterior insula and dorsal anterior cingulate cortex), dorsal attention network regions (frontal eye field and inferior parietal sulcus/superior parietal lobe), lateral prefrontal cortex (inferior and middle frontal gyri), sensorimotor regions (pre- and postcentral gyrus and supplementary motor area), and visual cortex. Extensive negative associations with RT were observed in default-mode network regions, including the medial prefrontal cortex, posterior cingulate, precuneus, angular gyrus, bilateral middle and inferior temporal gyrus, and medial temporal lobe. Comparing between- and within-subjects analysis results in brain areas revealed that several areas had opposite signs between the 2 levels of analysis (Fig. 3c). These divergent patterns underscore that group-level associations substantially mischaracterize the neural dynamics governing processing speed at the individual level. Taken together, these analyses provide direct empirical evidence of nonergodicity in brain-behavior relationships for RT.

## Nonergodic brain-behavior associations using latent model-based measures

To probe whether interpretable cognitive measures inferred by PRAD also exhibited nonergodicity, we related SSRT, probability of proactivity, and proactive delaying to brain activity at both levels of analysis.

Between subjects, SSRT showed widespread negative correlations with brain activity, including in frontal, parietal, and temporal areas, and in regions implicated in cognitive control (Fig. 4a). This suggests that individuals with faster inhibitory responses (lower SSRT) show greater activation in these regions during successful stopping, replicating previous findings[47]. In contrast, within subjects, SSRT showed

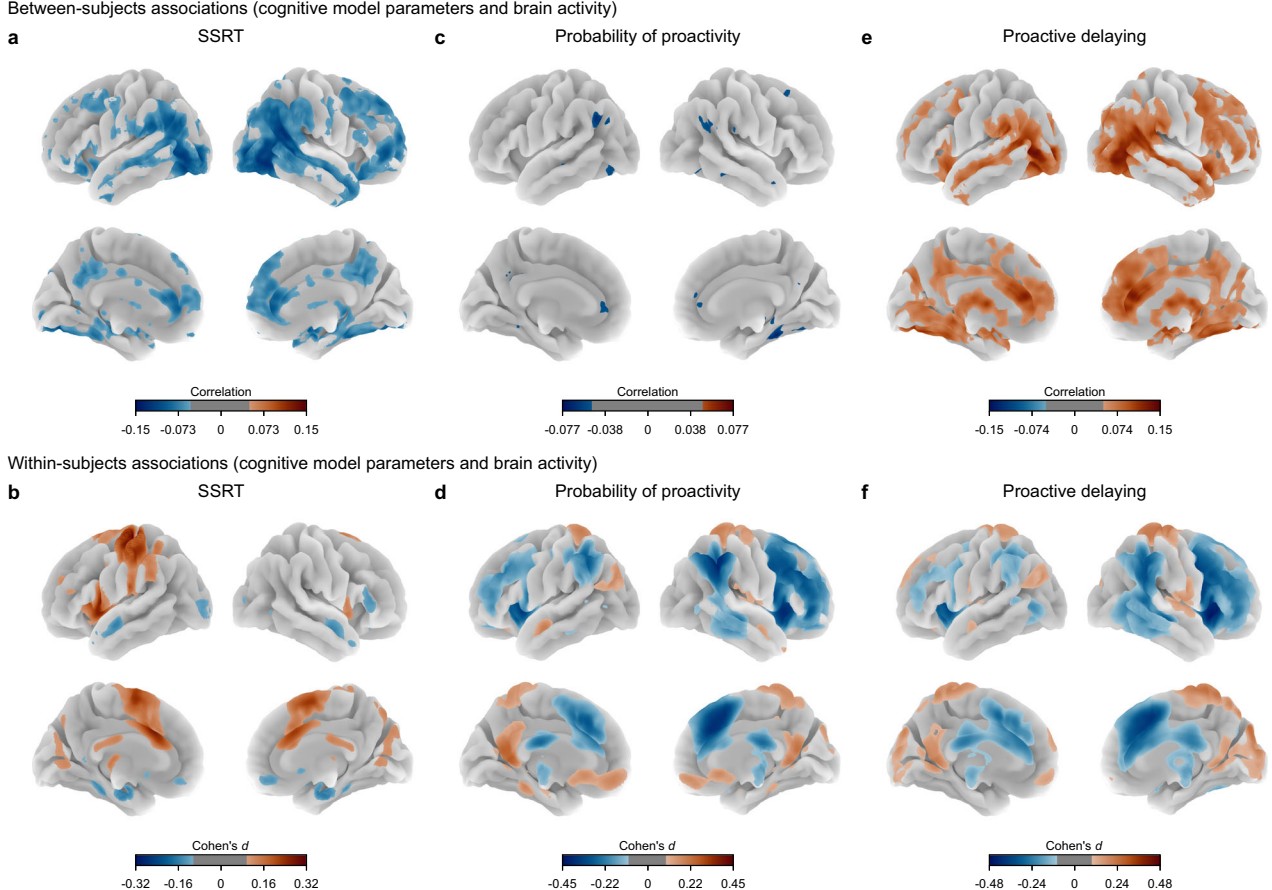

**Fig. 4 | Nonergodicity is exhibited by computational model parameters.**
**a**, **c**, **e** Between-subjects analysis. Whole-brain correlation maps of associations between subject-average brain activation (correct stop versus correct go activation) and cognitive model parameters: SSRT (**a**), probability of proactivity (**c**), and proactive delaying (**e**). Thresholded at Pearson $r \geq 0.05$. **b**, **d**, **f** Within-subjects analysis. Whole-brain Cohen's $d$ maps of associations between trial-by-trial brain activity and cognitive model parameters: SSRT (**b**), probability of proactivity (**d**), and proactive delaying (**f**). Thresholded at Cohen's $d \geq 0.1$. For both between- and within-subjects analyses: SSRT and probability of proactivity $N = 4423$; proactive delaying $N = 4137$. Between- and within-subjects associations for the model parameters provide further evidence that brain-behavior relationships underlying inhibitory control are nonergodic.

positive associations with brain activity, particularly in frontal and parietal regions (Fig. 4b and SI Fig. S2b). This suggests that on trials where individuals have slower inhibitory responses, they show increased activation in these areas.

Between subjects, probability of proactivity showed minimal correlations with brain activity (Fig. 4c). In contrast, within subjects, probability of proactivity showed negative associations in parietal, temporal, and lateral frontal regions, and positive associations in default mode network regions (Fig. 4d and SI Fig. S2c).

Between subjects, proactive delaying showed widespread positive correlations with brain activity, particularly in frontal and parietal regions (Fig. 4e). This indicates that individuals who engage in more proactive delaying exhibit higher activation in these areas during successful stopping. In contrast, within subjects, proactive delaying showed negative associations in frontal and parietal cortex (Fig. 4f and SI Fig. S2d).

These findings reveal a striking divergence of between-subjects and within-subjects brain-behavior relationships in inhibitory control. The reversal of association directions, particularly for SSRT and proactive delaying, suggests that inferences about the neural mechanisms underlying inhibitory control differ between the group and individual levels. This nonergodic pattern highlights the importance of considering both levels of analysis to more fully understand the neurocognitive dynamics of inhibitory control.

## Network-level visualization of between- and within-subjects brain-behavior associations

To elucidate the patterns of brain-behavior relationships across different functional brain networks, we visualized the between-subjects and within-subjects associations for SSRT, probability of proactivity, and proactive delaying. We used the Shirer network atlas[54] for our primary analysis as it includes the basal ganglia, a subcortical system important for the implementation of inhibitory control[31,55].

For SSRT (Fig. 5a), between-subjects analysis showed consistent negative correlations in most networks, with effects in the posterior salience, frontoparietal, and default mode networks (all $P_{FDR} < 0.01$); in contrast, within-subjects analysis revealed mixed negative and positive associations, with opposite (positive) associations in the posterior salience, precuneus, visuospatial, auditory, and sensorimotor networks (all $P_{FDR} < 0.01$). The reversal of association directions underscores the nonergodic nature of SSRT-related brain activity.

For probability of proactivity (Fig. 5b), between-subjects analysis demonstrated no significantly nonzero correlations in the networks except for a negative correlation in the language network ($P_{FDR} < 0.01$); however, within-subjects analysis unveiled a more complex pattern, with negative associations in the salience and frontoparietal networks and positive associations in the default mode network (all $P_{FDR} < 0.01$). This disparity highlights the importance of examining within-subjects dynamics for proactivity.

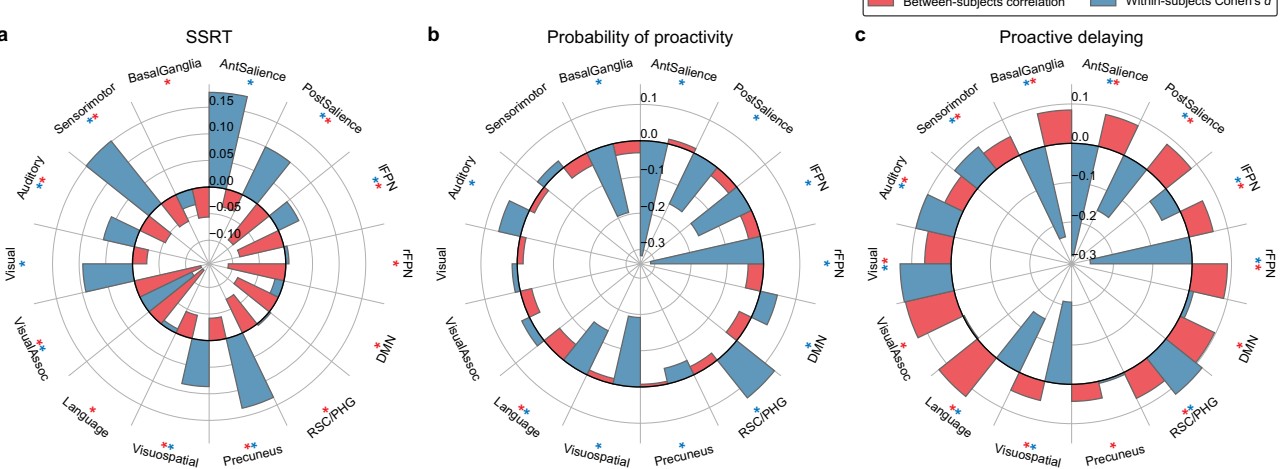

**Fig. 5 | Network-level comparison of between-subjects and within-subjects brain-behavior associations in inhibitory control.** The between-subjects analysis correlated subject-average brain activation (correct stop versus correct go activation) and cognitive model parameters: SSRT (**a**), probability of proactivity (**b**), and proactive delaying (**c**). The within-subjects analysis regressed trial-by-trial brain activity on cognitive model parameters: SSRT (**a**), probability of proactivity (**b**) and proactive delaying (**c**). Effect sizes are shown for both analyses (between subjects: Pearson $r$; within subjects: Cohen's $d$). Statistical significance is indicated by colored asterisks: red for between subjects ($P_{FDR} < 0.01$) and blue for within subjects ($P_{FDR} < 0.01$). Differences in the existence and direction of associations between between-subjects and within-subjects analyses were observed across multiple brain networks, including the anterior and posterior salience, left and right frontoparietal, and default mode networks. Two-sided permutation tests were used with FDR correction. SSRT between-subjects $P_{FDR}$: AntSalience = 0.022, Post-Salience < 0.001, lFPN < 0.001, rFPN < 0.001, DMN < 0.001, RSC/PHG < 0.001, Pre-cuneus = 0.0079, Visuospatial = 0.0038, Language < 0.001, VisualAssoc < 0.001, Visual = 0.072, Auditory < 0.001, Sensorimotor = 0.0014, BasalGanglia < 0.001. SSRT within-subjects $P_{FDR}$: AntSalience < 0.001, PostSalience < 0.001, lFPN =

0.0087, rFPN = 0.76, DMN = 0.31, RSC/PHG = 0.82, Precuneus < 0.001, Visuospatial < 0.001, Language = 0.65, VisualAssoc < 0.001, Visual<0.001, Auditory < 0.001, Sensorimotor < 0.001, BasalGanglia = 0.056. Probability of proactivity between-subjects $P_{FDR}$: AntSalience = 0.41, PostSalience = 0.11, lFPN=0.060, rFPN = 0.034, DMN = 0.050, RSC/PHG = 0.15, Precuneus = 0.63, Visuospatial = 0.38, Language = 0.0028, VisualAssoc=0.060, Visual = 0.37, Auditory = 0.37, Sensorimotor = 0.050, BasalGanglia = 0.060. Probability of proactivity within-subjects $P_{FDR}$: AntSa-lience < 0.001, PostSalience < 0.001, lFPN < 0.001, rFPN < 0.001, DMN = 0.0020, RSC/PHG < 0.001, Precuneus = 0.0056, Visuospatial < 0.001, Language < 0.001, VisualAssoc = 0.13, Visual = 0.35, Auditory < 0.001, Sensorimotor = 0.13, BasalGan-glia < 0.001. Proactive delaying between-subjects $P_{FDR}$: AntSalience < 0.001, Post-Salience < 0.001, lFPN < 0.001, rFPN < 0.001, DMN < 0.001, RSC/PHG < 0.001, Precuneus = 0.0048, Visuospatial = 0.0013, Language < 0.001, VisualAssoc < 0.001, Visual<0.001, Auditory = 0.0010, Sensorimotor = 0.0013, BasalGanglia < 0.001. Proactive delaying within-subjects $P_{FDR}$: AntSalience < 0.001, PostSalience < 0.001, lFPN = 0.0018, rFPN < 0.001, DMN = 0.55, RSC/PHG < 0.001, Precuneus = 0.81, Visuospatial < 0.001, Language < 0.001, VisualAssoc = 0.81, Visual < 0.001, Audi-tory < 0.001, Sensorimotor < 0.001, BasalGanglia < 0.001.

For proactive delaying (Fig. 5c), between-subjects analysis revealed positive correlations in all the networks (all $P_{FDR} < 0.01$); yet, within-subjects analysis showed predominantly negative associations, notably in the salience and frontoparietal networks (all $P_{FDR} < 0.01$). The retrosplenial cortex and parahippocampal gyrus nodes of the ventral default mode network showed a positive within-subjects association ($P_{FDR} < 0.01$).

The consistent reversals observed, for both reactive and proactive control measures, show that brain-behavior relationships in inhibitory control are not only nonergodic but also exhibit Simpson's paradox. A detailed characterization of the degree and location in the brain of Simpson's paradox can be found in the Supplementary Information (SI Fig. S3).

These network-level visualizations emphasize the divergent patterns between group-level and individual-level associations across different functional brain networks. Our findings indicate that both between-subjects and within-subjects perspectives are needed to comprehensively understand the neural dynamics underlying inhibitory control.

### Within-subjects analysis provides insights into dissociated reactive and proactive representations

Building on our findings of nonergodic brain-behavior relationships, we sought to deepen our understanding of how the brain implements proactive and reactive control at the individual level. While the preceding results demonstrate the importance of within-subjects analyses, they leave unresolved the question of how representations of reactive and proactive processes relate to each other within brain

networks. To investigate this, we used representational similarity analysis to examine the overlap between brain representations of reactivity (SSRT) and proactivity (probability of proactivity and proactive delaying) within individuals. For each subject and each brain network, we computed pairwise Pearson correlations between the subject's brain maps of SSRT and probability of proactivity, SSRT and proactive delaying, and probability of proactivity and proactive delaying, over the voxels in each network (Fig. 6a).

SSRT showed low similarity with both proactive measures (probability of proactivity and proactive delaying) in all networks, with median correlations ranging from −0.07 to 0.06. In contrast, the 2 proactive measures (probability of proactivity and proactive delaying) exhibited high similarity in all networks, with median correlations ranging from 0.61 to 0.67. In each network, the 3 similarity measures (SSRT and probability of proactivity, SSRT and proactive delaying, and probability of proactivity and proactive delaying) were significantly different from each other in their median values (Fig. 6b) (all $P_{FDR} < 0.01$). These findings suggest that representations of reactivity and proactivity are largely dissociated. This dissociation persists across multiple brain networks, indicating a fundamental separation in how the brain encodes reactive and proactive control processes.

### Within-subjects analysis reveals that subgroups with distinct control strategies have distinct brain-behavior associations

To understand how our within-subjects associations related to between-subjects variation in cognitive and task strategies, we examined the within-subjects results between subgroups showing adaptive and maladaptive regulation of reactive and proactive behaviors. These

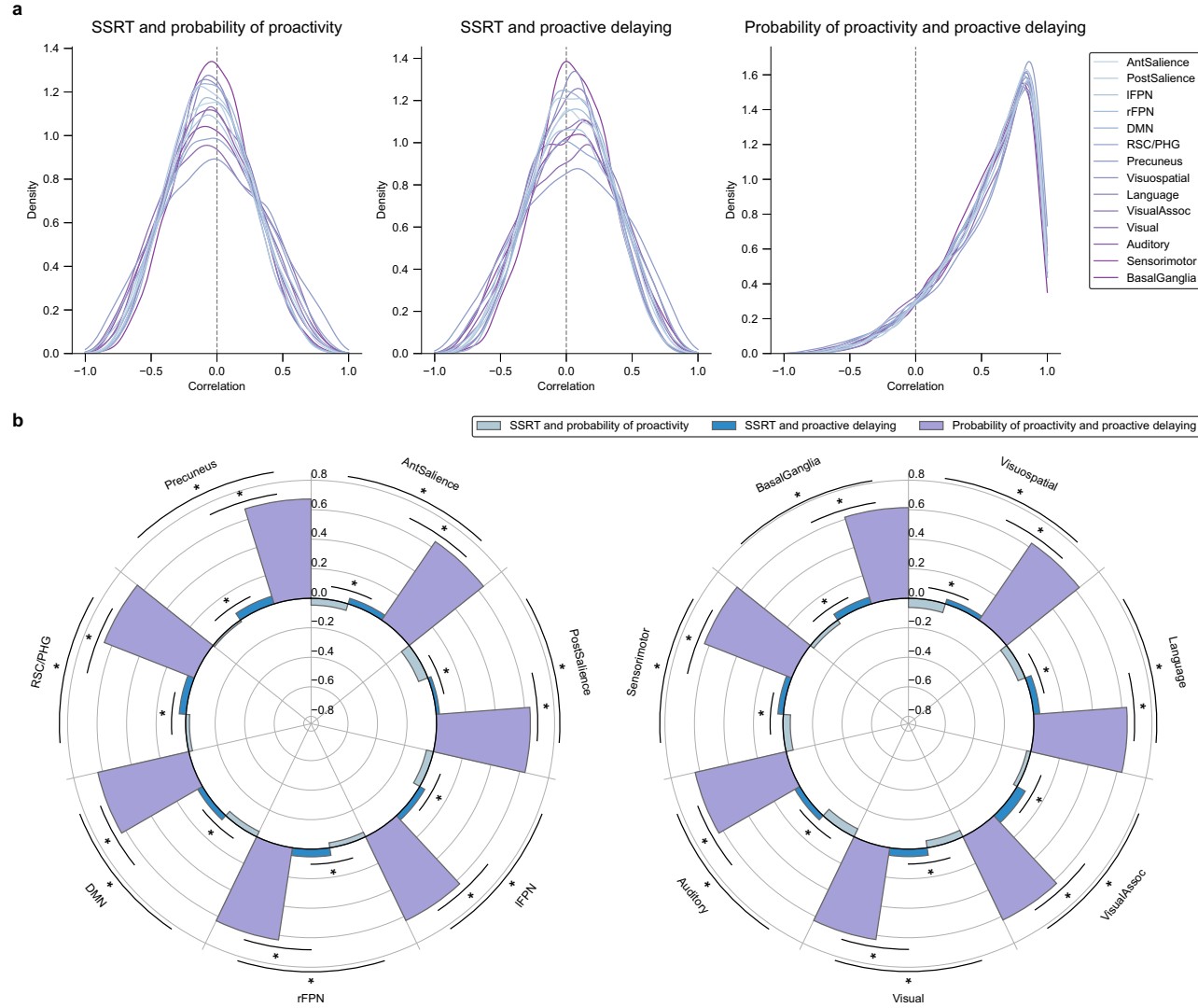

**Fig. 6 | Dissociated brain representations of reactive and proactive control processes.** For each subject, correlations were computed between pairs of within-subjects brain maps for the model parameters (SSRT, probability of proactivity, and proactive delaying) over each brain network. **a** Distributions of correlations over subjects. Kernel density estimation was performed. **b** Average correlations over subjects. The height of the bars is the median. Statistical significance of difference in medians is indicated by asterisks (* $P_{FDR}$ < 0.01). Reactive (SSRT) and proactive

(probability of proactivity, proactive delaying) control processes showed dissociated representations across all brain networks. This dissociation suggests that the brain employs distinct neural resources for reactive and proactive aspects of inhibitory control. In contrast, the two proactive measures showed high similarity, validating their representation of related cognitive processes. Two-sided permutation tests were used with FDR correction. All $P_{FDR}$ < 0.001.

subgroups were identified using PRAD model parameters $\gamma_1$ (governing regulation of stopping expectancy, and hence SSRT over trials) and $\theta_1$ (governing regulation of probability of proactivity over trials, in response to performance monitoring).

First, the cognitive model infers for each subject $\gamma_1$, which determines whether subjects adaptively ($\gamma_1 > 0$) or maladaptively ($\gamma_1 < 0$) regulate reactivity. Adaptive (maladaptive) regulation of reactivity involves increasing (decreasing) expectancy of a stop trial as the number of successive go trials increases. Since expectancy of a stop trial is one of several determinants of SSRT, $\gamma_1$ influences SSRT variation through time. Thus, we examined within-subjects associations between SSRT and brain activity separately among subjects with $\gamma_1 < 0$ ($N = 2487$) and $\gamma_1 > 0$ ($N = 1936$) (Fig. 7a). We found differences in the distributions of within-subjects SSRT associations between the $\gamma_1$ subgroups in all networks examined (all $P_{FDR} < 0.01$). The maladaptive regulation group had a larger (more positive) effect size in every network. In fact, across half of the networks, SSRT exhibited opposite associations with brain activity in the two subgroups. For example, in

the frontoparietal and default mode networks, SSRT displayed a positive association among subjects with $\gamma_1 < 0$ and a negative association among subjects with $\gamma_1 > 0$ (all $P_{FDR} < 0.01$). Moreover, this analysis revealed that some of the effects in the full sample were driven by subjects belonging to one of the subgroups. For example, in the anterior salience network, brain activity's positive association with SSRT in the full sample was driven by $\gamma_1 < 0$ subjects; among $\gamma_1 < 0$, the effect in the anterior salience had a Cohen's $d$ of $\sim 0.3$, twice that of the effect in the full sample, while among $\gamma_1 > 0$, there was no significant effect at all.

Second, the cognitive model infers for each subject $\theta_1$, which determines whether subjects adaptively ($\theta_1 < 0$) or maladaptively ($\theta_1 > 0$) regulate proactivity. Adaptive (maladaptive) regulation of proactivity involves increasing (decreasing) the probability of proactivity following a failed stop trial and decreasing (increasing) the probability of proactivity following a no-response go-trial. Therefore, $\theta_1$ influences the probability of proactivity's variation through time. Thus, we examined the probability of proactivity's within-subjects

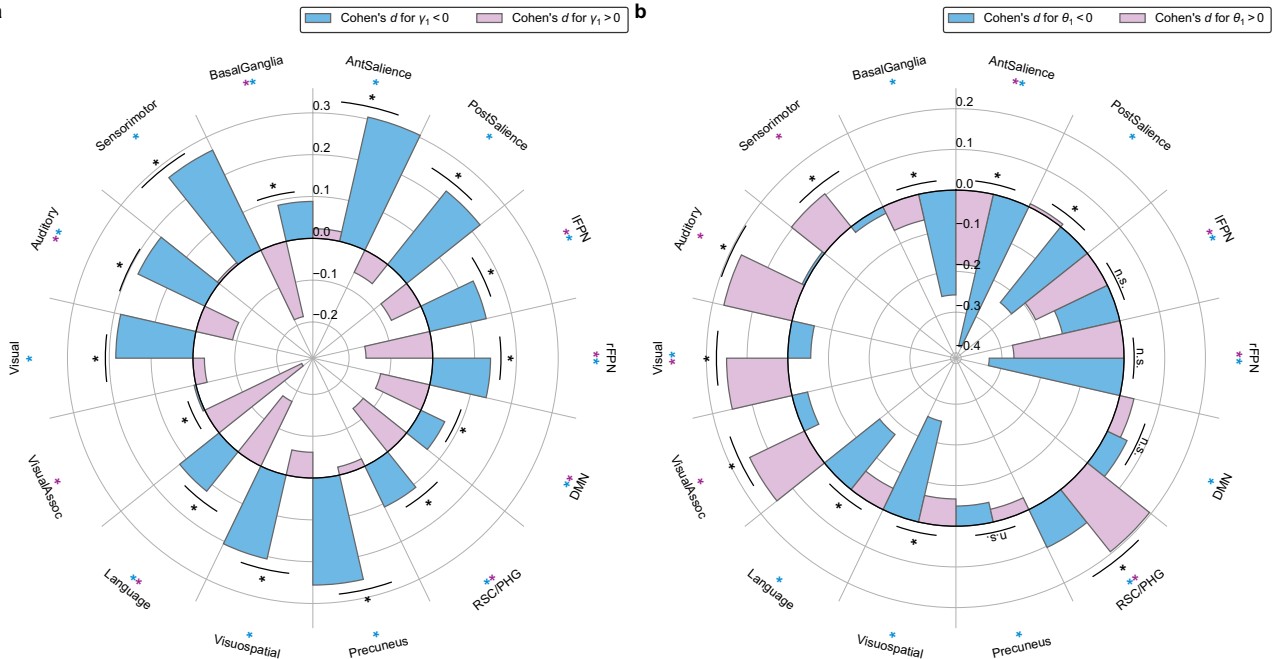

**Fig. 7 | Distinct brain-behavior associations for adaptive and maladaptive regulators of inhibitory control. a** Within-subjects associations of SSRT with brain activity between $\gamma_1$ subgroups. We identified two distinct subgroups of subjects with opposite profiles of attentional modulation. Subjects showed either maladaptive regulation ($\gamma_1 < 0$, $N = 2487$) or adaptive regulation ($\gamma_1 > 0$, $N = 1936$) of their expectancy of stopping over time. The two groups showed differences in within-subjects associations between SSRT and brain activity in various networks. This demonstrates that individual differences in attentional dynamics play a role in shaping the relationship between neural activity and inhibitory control processes. A colored asterisk indicates that a network or region's associations were nonzero among subjects with $\gamma_1 < 0$ (blue * $P_{FDR} < 0.01$) or among subjects with $\gamma_1 > 0$ (purple * $P_{FDR} < 0.01$). **b** Within-subjects associations of probability of proactivity with brain activity between $\theta_1$ subgroups. We identified two distinct subgroups of subjects with opposite profiles of performance monitoring. Subjects showed either adaptive regulation ($\theta_1 < 0$, $N = 3018$) or maladaptive regulation ($\theta_1 > 0$, $N = 1405$) of their proactivity over time. The two groups showed differences in within-subjects associations between probability of proactivity and brain activity in various networks. This demonstrates that the neural correlates of proactive control are influenced by an individual's strategy for adjusting proactivity in response to task outcomes. A colored asterisk indicates that a network or region's associations were nonzero among subjects with $\theta_1 < 0$ (blue * $P_{FDR} < 0.01$) or among subjects with

$\theta_1 > 0$ (purple * $P_{FDR} < 0.01$). For both panels, a black asterisk indicates that associations had different distributions between the two subgroups (* $P_{FDR} < 0.01$, n.s. $P_{FDR} \geq 0.01$). Two-sided permutation tests were used with FDR correction. For $\gamma_1 < 0$, $P_{FDR}$: AntSalience < 0.001, PostSalience < 0.001, lFPN < 0.001, rFPN < 0.001, DMN = 0.0015, RSC/PHG < 0.001, Precuneus < 0.001, Visuospatial < 0.001, Language<0.001, VisualAssoc = 0.82, Visual < 0.001, Auditory < 0.001, Sensorimotor < 0.001, BasalGanglia<0.001. For $\gamma_1 > 0$, $P_{FDR}$: AntSalience = 0.37, PostSalience = 0.014, lFPN = 0.0014, rFPN < 0.001, DMN < 0.001, RSC/PHG < 0.001, Precuneus = 0.43, Visuospatial = 0.012, Language<0.001, VisualAssoc<0.001, Visual = 0.29, Auditory < 0.001, Sensorimotor = 0.80, BasalGanglia < 0.001. For $\gamma_1 < 0$ versus $\gamma_1 > 0$, all $P_{FDR} < 0.001$. For $\theta_1 < 0$, $P_{FDR}$: AntSalience < 0.001, PostSalience < 0.001, lFPN < 0.001, rFPN < 0.001, DMN = 0.0059, RSC/PHG < 0.001, Precuneus = 0.0079, Visuospatial < 0.001, Language < 0.001, VisualAssoc = 0.042, Visual = 0.0031, Auditory = 0.73, Sensorimotor = 0.36, BasalGanglia < 0.001. For $\theta_1 > 0$, $P_{FDR}$: AntSalience < 0.001, PostSalience = 0.77, lFPN < 0.001, rFPN < 0.001, DMN = 0.21, RSC/PHG < 0.001, Precuneus = 0.24, Visuospatial = 0.018, Language=0.035, VisualAssoc < 0.001, Visual <0.001, Auditory < 0.001, Sensorimotor < 0.001, BasalGanglia=0.021. For $\theta_1 < 0$ versus $\theta_1 > 0$, $P_{FDR}$: AntSalience<0.001, PostSalience < 0.001, lFPN = 0.079, rFPN = 0.18, DMN = 0.63, RSC/PHG = 0.0031, Precuneus = 0.65, Visuospatial < 0.001, Language = 0.0012, VisualAssoc < 0.001, Visual < 0.001, Auditory < 0.001, Sensorimotor < 0.001, BasalGanglia < 0.001.

associations with brain activity separately among subjects with $\theta_1 < 0$ ($N = 3018$) and $\theta_1 > 0$ ($N = 1405$) (Fig. 7b). We found consistent differences in the probability of proactivity's within-subjects associations between these subgroups. Generally, subjects with adaptive regulation ($\theta_1 < 0$) had more negative associations between probability of proactivity and brain activity. This negative coupling between proactivity and brain activity was pronounced within the adaptive regulation group in the anterior salience network (Cohen's $d$ of $\sim 0.4$). This analysis also clarified how adaptive and maladaptive regulation of proactivity contributed to the effects observed in the full sample. The within-subjects positive association between probability of proactivity and activation in the retrosplenial cortex and parahippocampal gyrus network was observed in both $\theta_1$ subgroups (both $P_{FDR} < 0.01$), but it was twice as large among subjects who maladaptively regulated proactivity.

These results demonstrate that population subgroups related to adaptive and maladaptive regulation of inhibitory control demonstrate different, and even opposite, within-subjects brain-behavior associations.

## Stability of within-subjects brain-behavior associations
To assess the stability of our within-subjects findings, we performed bootstrap resampling at varying sample sizes (Fig. 8). Within-subjects associations between brain activity and the model parameters (SSRT, probability of proactivity, and proactive delaying) were stable, even in modest sample sizes. In each of 5 independent sets of brain areas, resampled results showed high similarity with results in the full sample. Key findings were consistently observed in samples as small as 25 subjects, with some effects requiring larger samples to emerge reliably. This stability suggests the validity of our within-subjects approach to understanding neurocognitive mechanisms of inhibitory control. A detailed description of these results is in the Supplementary Information.

## Robustness of results to analytical choices
We conducted several control analyses to test the robustness of our findings of nonergodicity and of our inferred within-subjects relationships.

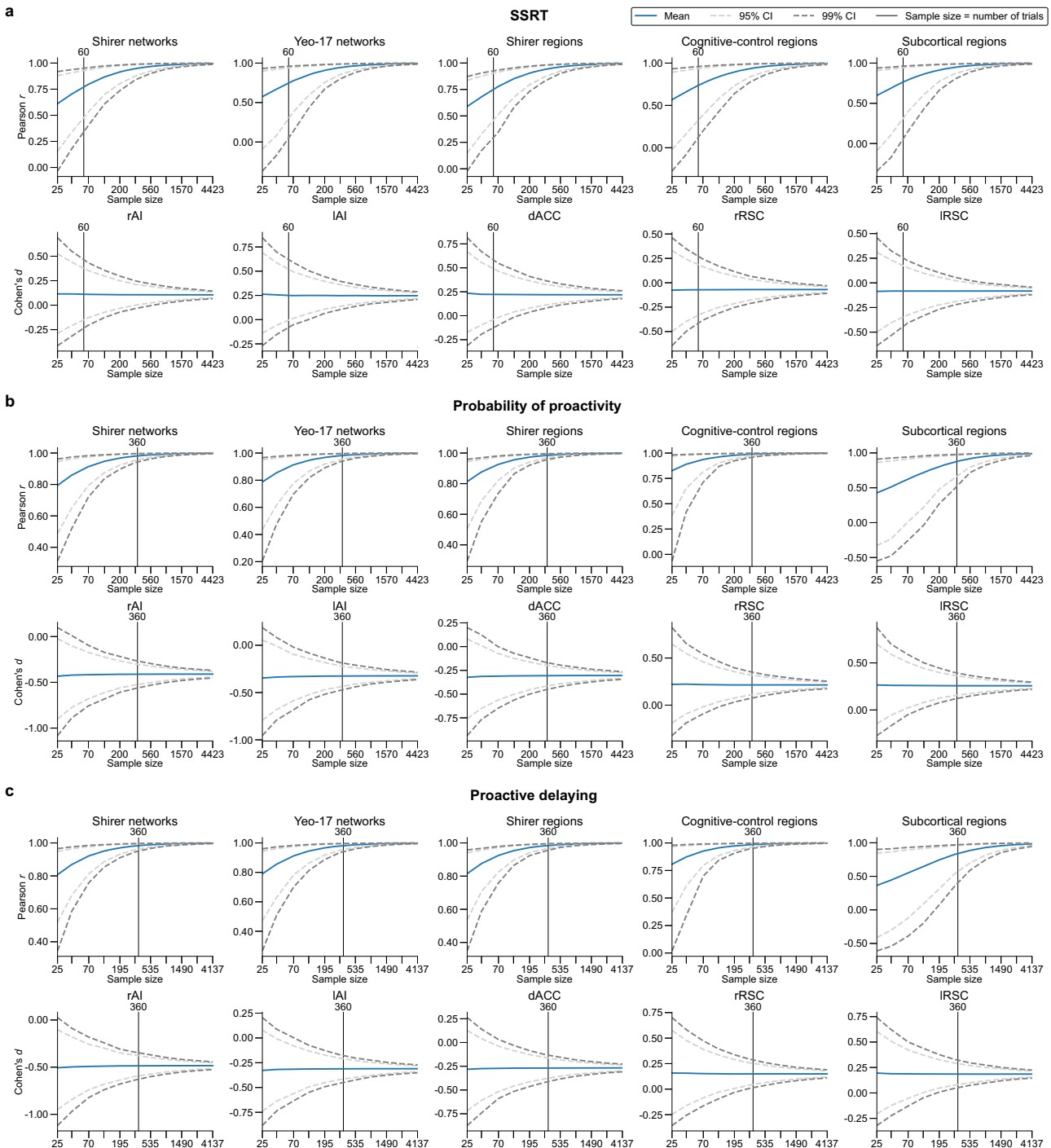

**Fig. 8 | Stability analysis of within-subjects associations. a–c** Stability plots for SSRT (**a**) probability of proactivity (**b**) and proactive delaying (**c**). In each panel, the top plots show the distributions of correlation between the effect in resamples and the effect in the full sample, while the bottom plots show the distributions of the effect in selected regions. For each plot, 10,000 resamples of *n* subjects were drawn with replacement for each sample size *n*, and Cohen's *d* was calculated for each resample. For top panel plots, correlations were then computed for each resample over the areas belonging to the set of brain areas. The dashed lines depict 95% and 99% bootstrap confidence intervals, and vertical black lines mark where sample size equals the number of trials used per subject to estimate the within-subjects association. Within-subjects associations demonstrated stability for all 3 cognitive model parameters across 5 different collections of brain areas and in 5 regions of interest, even at modest sample sizes. This stability supports the reliability of brain-behavior associations in inhibitory control processes.

We tested three alternative between-subjects approaches for go RT (SI Fig. S4) and the model parameters (SI Fig. S5), which assessed robustness to alternative subject-average measures of brain activation (task contrasts). In all, we examined 4 different strategies of performing between- and within-subjects analysis of how 1 observed behavioral measure, 1 latent SSRT measure, and 2 latent proactivity measures related to brain activity. In each of these 16 comparisons, within- and between-subjects inferences about brain-behavior associations diverged. This consistent finding across multiple analytical approaches and measures provides robust empirical evidence that the neurocognitive dynamics of inhibitory control are substantively nonergodic, meaning that within-subjects analysis provides

meaningfully different and novel inferences about how neurocognitive processes unfold in the brain at an individual level. A detailed description of these control analyses and their results is in the Supplementary Information.

In addition, we examined whether within-subjects brain-behavior associations based on PRAD parameters were sensitive to the PRAD model's assumptions (SI Fig. S6). We computed brain-behavior associations for behavioral measures from 3 control models: the PRAD model fit separately for each fMRI run, a version of the PRAD model that does not account for drift rate variability on go trials, and a model that assumes variability in trial-level SSRT but without the attentional and proactive mechanisms. Brain-behavior associations using each of these control models closely matched those using PRAD.

We also probed whether our average within-subjects brain-behavior associations for go RT, SSRT, probability of proactivity, and proactive delaying were reliably observed across SST fMRI runs (SI Fig. S7). We found high similarity between results from the first and second runs, ranging from Pearson $r = 0.80$ (SSRT) to $r = 0.98$ (go RT). And finally, we examined whether our group-aggregated within-subjects results were robust to the use of a 2 stage random effects meta-analysis rather than simple averages, finding very high similarity between these two aggregation approaches (all Pearson $r \geq 0.987$) (SI Fig. S8).

Overall, our observations of nonergodicity did not depend on the method of between-subjects analysis and our inferred within-subjects relationships were robust to changing various PRAD model assumptions, the use of only a single run of the task, and several methods of group aggregation.

## Discussion

Nonergodicity, the divergence between group-level and individual-level patterns, poses a fundamental challenge for cognitive neuroscience. The central question we addressed is whether empirical brain-behavior relationships underlying cognitive control are nonergodic, and critically, whether any nonergodicity substantively challenges our understanding of inhibitory control mechanisms based on traditional between-subjects analysis. Leveraging a large community sample and a novel dynamic computational model[50], we examined neurocognitive dynamics at both between-subjects and within-subjects levels.

Our findings reveal pervasive and substantive nonergodicity in brain-behavior relationships underlying cognitive control. This nonergodicity was observed across multiple measures—from observed RT to latent model-derived measures of reactive and proactive control—and manifested as Simpson's paradox[4,10,12,18]. Beyond demonstrating nonergodicity, our within-subjects analyses suggest mechanistic perspectives unavailable from conventional approaches. We found that reactive and proactive control processes have orthogonal neural representations, and that individual differences in cognitive strategies are associated with distinct within-subjects brain-behavior relationships.

Results underscore the critical importance of distinguishing between-subjects and within-subjects inferences about the brain—a paradigm we call "nonergodicity neuroscience." Our findings challenge the implicit assumption of ergodicity in cognitive neuroscience and demonstrate that examining how brain activity and behavior co-vary within individuals over time provides fundamentally different insights than examining how they co-vary across individuals.

Before examining brain-behavior relationships using model-derived latent parameters, we first investigated whether nonergodicity could be seen using a directly observable behavioral measure. We examined associations between brain activity and go RT at both between-subjects and within-subjects levels. This approach provides a straightforward exploration of nonergodicity that does not depend on computational modeling assumptions.

Between subjects, RT-activation associations were predominantly positive and relatively sparse. In stark contrast, within subjects, trial-to-trial RT fluctuations showed extensive and bidirectional coupling with brain activity across multiple networks. Notably, the default mode network (which includes medial prefrontal cortex, posterior cingulate/precuneus, and angular gyrus) showed a negative association with RT within subjects (greater suppression on slower trials) but a positive association between subjects. Additionally, many networks (e.g., salience, visuospatial, and frontoparietal networks) showed positive or negative within-subjects associations but null effects between subjects.

The striking divergence between analysis levels, with sparse positive between-subjects associations contrasting against extensive bidirectional within-subjects coupling, demonstrates that group-level patterns fail to capture the rich neural dynamics underlying moment-to-moment performance variations. Most notably, the reversal of associations in default mode network regions exemplifies Simpson's paradox: these regions showed a positive relationship with RT between individuals, yet within individuals, they showed negative trial-by-trial coupling with RT. This pattern underscores the critical need for trial-resolved, subject-specific analyses to accurately characterize the neural mechanisms governing processing speed.

These findings establish that nonergodicity in brain-behavior relationships is observable even with simple, directly measured behavioral variables. This demonstration provides important context for interpreting the model-based analyses that follow. We show that the divergence between group-level and individual-level patterns is not an artifact of computational modeling but reflects a fundamental property of neurocognitive dynamics. Having established this foundation, we next turn to examining nonergodicity using model-derived measures of proactive and reactive control, which allow us to dissect inhibitory control into its constituent cognitive processes.

While the analysis of overt behavior (go RT) provides compelling evidence for nonergodicity, using RT for within-subjects analysis of this task has important limitations. First, RT is unavailable on approximately half of stop trials (successful stops), limiting our ability to examine brain-behavior relationships across the full task. Second, RT does not directly capture inhibitory control capabilities, the central focus of the SST. Third, observed RT conflates multiple neurocognitive processes that combine to produce behavior, including metacognitive processes such as performance monitoring and adjustment of proactive control strategies. To overcome these limitations and provide a more refined examination of nonergodicity, we leveraged latent dynamic measures from the PRAD model[50].

The PRAD model provides trial-level estimates of distinct cognitive processes underlying inhibitory control: SSRT (reactive control), probability of proactivity, and proactive delaying (both reflecting proactive control). By dissociating these elemental processes, the model also allows us to examine whether nonergodicity manifests differently across distinct components of cognitive control and to reveal mechanisms that would be obscured when examining only observed behavior.

Our findings reveal pervasive nonergodic patterns across multiple measures of inhibitory control. For reactive control using PRAD-derived SSRT, we observed a striking reversal between analysis levels. Within subjects, longer trial-level SSRTs were positively associated with brain activity in the anterior and posterior salience networks. This suggests that poorer reactive control on a given trial is accompanied by greater neural engagement, potentially reflecting compensatory mechanisms or increased cognitive resource demands when individuals struggle to inhibit responses. In stark contrast, between subjects, we observed no significant association in the anterior salience network and a negative association in the posterior salience network.

For proactive control, within-subjects analysis revealed negative associations between trial-level engagement of proactive control and

brain activity in the frontoparietal, salience, and subcortical systems. This pattern is consistent with a mechanistic account wherein greater proactive control reduces the need to engage cognitive control networks that facilitate reactive inhibition. This finding aligns with recent theoretical frameworks proposing that proactive control modulates reactive control via preparatory processes[56], providing empirical evidence for this mechanism at the neural level within individuals.

Surprisingly, we also observed positive associations between trial-level proactive control and default mode network activity, including the posterior medial cortex and the ventromedial prefrontal cortex—the two core cortical nodes that anchor the default mode network[57]. This pattern may reflect internally oriented processing that supports proactive regulation, such as anticipating upcoming events or maintaining task goals. This finding challenges the traditional view of the DMN as a "task-negative" network and suggests a more critical role in supporting aspects of cognitive control. In sharp contrast, between-subjects analysis failed to capture these dynamic relationships, showing minimal to no association between one measure of proactive control and brain activations.

The divergence between within-subjects and between-subjects patterns for both reactive and proactive control provides further evidence for nonergodicity in human cognitive neuroscience and underscores a fundamental limitation of conventional approaches. Between-subjects analyses may not capture the complex and dynamic nature of proactive and reactive control processes as they unfold within individuals over time. Our findings suggest that nonergodicity neuroscience approaches are crucial for understanding neurocognitive functions at the individual subject level. This has important implications for understanding cognitive control across development and in psychopathology, where within-individuals dynamics may be particularly important for identifying mechanisms of change and targets for intervention.

Having established nonergodicity, we sought to further elucidate the neural architecture underlying inhibitory control and clarify the relationship between reactive and proactive control mechanisms. We employed representational similarity analysis, a powerful method for investigating the informational content of brain activity patterns[58]. This approach allows us to compare the similarity of neural representations across different cognitive processes, providing insights into how the brain organizes and processes information[59]. In our study, we used representational similarity analysis to examine the overlap between brain representations of reactivity (SSRT) and proactivity (probability of proactivity and proactive delaying) within individuals. By comparing representational patterns across different cognitive processes, we sought to determine whether reactive and proactive control processes rely on shared or distinct neural resources.

Our analysis revealed a striking within-subjects dissociation between the neural representations of reactive and proactive control processes. Across all examined brain networks, we found low similarity between representations of SSRT and representations of each proactive measure (probability of proactivity and proactive delaying). In contrast, the two proactive measures showed high similarity with each other. This pattern is consistent with reactive and proactive control processes being represented orthogonally in the brain. These measures of proactivity affecting the go process and reactivity affecting the stopping process are explicitly defined by the PRAD model in the standard SST, in contrast to measures based on modified versions of the standard task that attempt to experimentally induce proactive and reactive aspects of inhibitory control on different trials[36,60–62].

Theoretically, our findings suggest that reactive and proactive processes are implemented through distinct neural mechanisms[52,63]. Given that our study examined children, the clear separation of reactive and proactive representations may reflect a developmental stage in the organization of cognitive control processes. The separation we found may allow for independent development of reactive and

proactive strategies, potentially explaining individual differences in inhibitory control abilities[52]. Future studies could investigate whether this orthogonality persists or changes with age[64].

A key advantage of our dynamic computational modeling approach is the ability to identify meaningful subgroups based on individual differences in cognitive strategies, allowing us to investigate whether the nature of nonergodic patterns varies across individuals. Our behavioral investigation revealed two model parameters that control within-subjects variability in inhibitory control[50]: $\gamma_1$ representing individual differences in sustained attentional modulation of trial-level stopping expectancy, and $\theta_1$ representing individual differences in performance monitoring and regulation of proactive control. These parameters distinguish individuals who adaptively versus maladaptively regulate their inhibitory control. We used these parameters to stratify participants and examined whether within-subjects brain-behavior associations differed between adaptive and maladaptive regulators. This revealed striking heterogeneity in neural mechanisms underlying inhibitory control.

For reactive control ($\gamma_1$ subgroups), individuals who adaptively versus maladaptively regulated stopping expectancy showed distinct within-subjects associations between trial-level SSRTs and brain activity, with different, and often opposite, patterns across most brain networks. Notably, the association between anterior salience network activation and poorer trial-level reactive control, observed in the full sample, was driven entirely by subjects who maladaptively regulated reactivity. This suggests that the neural signature of reactive control failures differs fundamentally between individuals depending on their attentional regulation strategies.

For proactive control ($\theta_1$ subgroups), individuals who adaptively versus maladaptively regulated proactivity showed different within-subjects associations in most networks. The maladaptive regulation group showed weaker proactivity-related suppression of the anterior salience network and stronger proactivity-related activation in the retrosplenial cortex and parahippocampal gyrus, regions anchoring the ventral posterior aspects of the default mode network. This pattern suggests that maladaptive regulators may rely more heavily on internally oriented processing when engaging proactive control, potentially at the expense of efficient proactive control implementation.

These findings have important implications for understanding individual differences in cognitive control. First, they demonstrate that adaptive and maladaptive regulation strategies are associated with substantively different neural implementations of inhibitory control. Second, the heterogeneity revealed by subgroup analyses underscores a critical limitation of conventional group-level analyses, which fail to show this important variability by averaging over it. Third, these results highlight the importance of considering metacognitive processes, attentional modulation and performance monitoring, when studying inhibitory control. By leveraging dynamic computational models to identify interpretable subgroups, we can begin to understand not just whether brain-behavior relationships are nonergodic, but how brain-behavior relationships vary across individuals with different cognitive profiles.

Leveraging the large-scale ABCD dataset, we addressed the critical challenge of replicability in human neuroscience[65,66]. Our analyses revealed that within-subjects associations were stable and reliably detectable, even in sample sizes typical of cognitive neuroscience studies. Bootstrap resampling analyses showed the reliability of key findings across different sample sizes. For instance, the association between proactivity measures and right anterior insula suppression was consistently observed in over 95% of samples, even with sample sizes of $N = 25$.

Moreover, our findings of nonergodicity were robust to various analytical strategies and modeling choices. When comparing the results of various between- and within-subjects approaches, there were variations in the details of brain-behavior inferences, but nonergodic dissociations persisted across every approach to brain-behavior

association, including the use of observed dynamic measures (go RT) rather than model-based latent dynamic measures.

By demonstrating that nonergodicity patterns are robust and detectable even in modest sample sizes, our study provides a foundation for future research into nonergodicity in brain function. It also suggests that meaningful insights into neurocognitive mechanisms can be gained from studies with more typical sample sizes, although larger samples provide greater precision and the ability to detect subtler effects. The stability and robustness of our findings suggest nonergodicity's applicability to diverse research and clinical contexts, including understanding cognitive processes related to inhibitory control and studying psychiatric disorders[67].

Our study provides evidence for pervasive nonergodicity in the neurocognitive processes underlying inhibitory control. Using a large, community-representative sample from the ABCD study and combining task fMRI data with a dynamic computational model, we demonstrate that brain-behavior relationships differ fundamentally when examined at group versus individual levels. Critically, we found that group-level patterns not only disagreed with individual-level patterns but, in many cases, reversed them, exemplifying Simpson's paradox at the neural level. Our results challenge the implicit assumption of ergodicity that pervades cognitive neuroscience research, and they invite neuroscientists to reflect on their questions and on their methods—between or within individuals?

Our work also has significance beyond neuroscientific methodology.

The results advance our understanding of cognitive control. Within-subjects analyses revealed insights entirely unavailable from conventional approaches: dissociated neural representations of proactive and reactive control, dynamic interplay between cognitive control and default mode network areas, and systematic variation in neural mechanisms across individuals with different strategies. These findings, moreover, highlight an important direction for nonergodicity neuroscience: developing dynamic computational models capable of identifying elemental cognitive processes at the trial level.

The findings also have implications for medicine. Neurobiological features that distinguish diagnostic groups may differ fundamentally from the mechanisms driving symptom fluctuations within individuals or predicting treatment response. Personalized interventions targeting cognitive control must be tailored to individual-specific neural dynamics rather than extrapolated from group-average patterns.

More broadly, appreciating the nonergodic nature of neurocognitive processes is essential for advancing our understanding of human cognition in health, development, and disease, and may have important implications for interpreting artificial neural networks.

## Methods

### Inclusion criteria

Data were from the baseline visit of the ABCD study[25] (Collection #2573), $N = 11817$. Approval was received from institutional review boards at the University of California San Diego and the study sites. Informed consent was obtained from all participants. The participants were 9 or 10 years old. Both the children and their guardians were compensated.

Subjects were excluded if they did not meet each of the following criteria: meet the ABCD study's SST task-fMRI inclusion recommendations (in abcd_imgincl01.txt, imgincl_sst_include==1; $N = 3546$ excluded); have 2 SST fMRI runs of good quality (in mriqcrp20301.txt, iqc_sst_total_ser==iqc_sst_good_ser==2; $N = 677$ excluded); are successfully fit with the cognitive model of the SST ($N = 562$ excluded); have enough volumes acquired to cover the SST experiment (the last SST trial must have happened no more than 2 seconds after the final volume was acquired; $N = 5$ excluded); have mean framewise displacement of less than 0.5 mm for both runs (calculated using the method of ref. 68.; $N = 2058$ excluded); have release 4.0 minimally

processed events.tsv files of shape (181,3) for both runs ($N = 7$ excluded); and have consistent release 4.0 behavioral data (in release 4.0, for some subjects, the "sst.csv" files from ABCD Task fMRI SST Trial Level Behavior, abcd_sst_tlb01, disagreed with the minimally processed "events.tsv" files; for example, one trial might be labeled a go trial by one file and a stop trial by the other; $N = 102$ excluded). Then, we excluded siblings by randomly keeping one member from each family (using the genetic_paired_subjectid variables from gen_y_pihat; $N = 426$ excluded) and excluded subjects without scanner serial number recorded (in mri_y_adm_info, missing mri_info_deviceserialnumber; $N = 11$ excluded). Applying these inclusion criteria left us with a sample of $N = 4423$. For analyses involving the proactive delaying, a further 286 subjects were excluded who had no trials with probability of proactivity greater than 0.5 during at least one run, and therefore, by definition, a proactive delaying of 0 for all trials of at least one run. For these subjects, we were unable to examine within-subjects relationships between proactive delaying and brain activity. To maintain comparability of the between- and within-subjects analyses, we also excluded these subjects from the between-subjects analyses involving proactive delaying. Thus, analyses involving the proactive delaying used a sample of $N = 4137$. For the 4423 subjects, sex assigned at birth was male for 2111 and female for 2312.

### Brain imaging

Imaging acquisition for the ABCD SST is detailed in other work[69]. We used the minimally processed data from ABCD release 4.0 (Collection #2573), which included distortion correction and motion correction[70]. We then further processed the images using Nilearn and FSL FLIRT: (1) initial volumes were removed (Siemens: 8, Philips: 8, GE DV25: 5, GE DV26 and other GE versions: 16); (2) the mean image in the time dimension was computed using mean_img from the Nilearn image module; (3) the mean image was registered to an echo-planar imaging template in MNI152 space (SPM12's toolbox/OldNorm/EPI.nii) using FSL FLIRT, and an affine of this transformation was obtained; (4) the time-series of images was spatially normalized to MNI152 space with the affine from the previous step using FSL FLIRT; and (5) the images were smoothed with a Gaussian filter with a full-width at half maximum of 6 mm using smooth_img from the Nilearn image module.

### Bayesian modeling of cognitive dynamics

The PRAD model[50] incorporates latent dynamics that respond to endogenous and exogenous variables, with trait measures governing the interaction of such endogenous and exogenous variables with latent processes, giving rise to non-stationary dynamics. Overall, the PRAD model incorporates separate evidence accumulation (drift-diffusion) processes for the go and stop processes, similar to a canonical horse-race model[71]. However, in addition to typical drift-diffusion process parameters, PRAD includes individual trait-like measures (such as persistence in belief updating, proclivity for proactive control strategies, degree of performance monitoring based modulation, and sensitivity to attentional modulation) and dynamic trial-level measures (such as trial level modulations of the probability of proactivity, proactive delays in responding, and SSRT). The dynamic trial-level measures are of primary interest in this study. The full PRAD model is specified below.

The go process is modeled as a drift diffusion process, with a trial-invariant non-decision time ($\tau_G$) and initial directional bias ($\beta_G$), but a trial-varying decision threshold ($\alpha_{G,t}$) and drift rate ($\delta_{G,t}$). The dynamic decision threshold

$$\alpha_{G,t} = \alpha_{G1}(1 - \epsilon_{C,t-1}) + \alpha_{G2}(\epsilon_{C,t-1}),$$

where ($\alpha_{G1}, \alpha_{G2}$) are distinct threshold levels and $\epsilon_{C,t-1}$ is an indicator of whether the left vs right choice on the previous trial was erroneous.

Thus, the dynamic threshold implements a form of performance monitoring and varies between two levels based on the outcome of the previous trials, with $\alpha_{G1}$ being the default threshold and $\alpha_{G2}$ reflecting the threshold after post-error adjustments. The dynamic drift rate

$$\delta_{G,t} = \frac{\delta_0 \, \mathbb{S}_{LR,t}}{\left(1 + e^{-12(stim_t - \kappa_0)}\right)} \text{ if } stim_t \neq 0, \text{ else } 0.$$

Here $\delta_0$ is a measure of the maximum drift rate for an individual, with the actual drift rate depending on the duration of the go stimulus ($stim_t$) and an individual parameter $\kappa_0$, which can be interpreted as the stimulus duration at which the drift rate is half the maximum. $\mathbb{S}_{LR,t}$ assumes values 1 or −1 depending on the direction of the go stimulus (left or right). PRAD allows for trial level changes to the drift rate, overcoming the issues with variable go stimulus durations highlighted in previous work[72]. We also tested a version of PRAD (PRAD-G) whose go drift rate is fixed on go trials, which is described in the Supplementary Information.

In addition, in the PRAD model, the onset of the go process may be deliberately delayed in anticipation of a stop signal. This dynamic adaptation is modeled by adding a further delay $\omega_t$ to the go process to reflect proactive delayed responding to the go stimulus, where

$$\omega_t = \lambda_t \, \rho_t.$$

Here, $\lambda_t$ reflects a trial-level belief updating process, based on the history of stop signal delays (SSD) encountered, and is an internal noisy estimate of the prospective anticipated SSD. The parameter $\mu$ ($0 < \mu < 1$) reflects persistence in belief updating, with high persistence implying a lower decay rate of older SSDs encountered. Further, letting $SSD_t$ be the SSD and $\mathbb{I}_{S,t}$ be a stop trial indicator,

$$\lambda_t = \frac{\left(\sum_{i=1}^{t-1} \mu^{t-i-1} \, SSD_i \, \mathbb{I}_{S,i}\right)}{\left(\sum_{i=1}^{t-1} \mu^{t-i-1} \, \mathbb{I}_{S,i}\right)}$$

$\rho_t$ is a binary variable representing cognitive state. $\rho_t$ indicates the presence ($\rho_t = 1$) or absence ($\rho_t = 0$) of a proactive cognitive state on trial $t$. The proactive delayed responding is only initiated on proactive cognitive states. Proactive cognitive states are governed by a baseline proclivity for proactivity ($\theta_0$), and a performance monitoring based modulation ($\theta_1$).

$$\rho_t \sim \text{Bernoulli}\left(\frac{1}{1 + e^{-(\theta_0 + \theta_1(\epsilon_{G,t-1}) - \theta_1(\epsilon_{S,t-1}))}}\right).$$

Here, $\epsilon_{G,t-1}$ is an indicator of a go-omission (incorrectly stopping on a go trial) on the previous trial, and $\epsilon_{S,t-1}$ is an indicator of a stopping error (not stopping on a stop trial). The PRAD model assumes that the correction in terms of increasing or decreasing the probability of a proactive cognitive state following these two types of trials will be in opposite directions. The sign of $\theta_1$ is an indicator of adaptivity or maladaptivity of the performance monitoring mechanism, and the absolute value of $\theta_1$ denotes the sensitivity of the state-switching mechanism to errors.

The stop process is modeled as a drift diffusion process with a trial-invariant non-decision time ($\tau_S$), decision threshold ($\alpha_S$), and drift rate ($\delta_S$), but a trial-varying bias ($\beta_{S,t}$). The stop process begins at the onset of the stop signal. The initial bias is

$$\beta_{S,t} = \left(\frac{1}{1 + e^{-\gamma_1(nSSD_t - \gamma_0)}}\right).$$

Here, $nSSD_t$ reflects the number of trials since a stop signal was last encountered. This reflects an attentional mechanism that modulates the stopping bias $\beta_{S,t}$ (which varies from 0 to 1). Positive values of $\gamma_1$ result in an increase in stopping bias as $nSSD_t$ increases. Similarly, negative values of $\gamma_1$ result in a decrease in stopping bias as $nSSD_t$ increases. The absolute value of $\gamma_1$ measures the sensitivity to attentional modulation. The $\gamma_0$ parameter is a measure of the value of $nSSD_t$ when stopping bias is neutral (0.5).

Both the go and stop processes are implemented within a hierarchical Bayesian modeling framework in JAGS[73], using the Wiener distribution[74], which produces a joint distribution of the RTs and the decision choice on each trial. The RTs of the go process correspond to the RTs for pressing the left or right buttons in response to the go stimulus. The RTs of the stop process correspond to the SSRT. The stop process is only initiated on stop trials after the appearance of the stop stimulus (which appears after a delay corresponding to the SSD). The SSRT is not manifested as a behavioral action. Rather, if the SSRT, which is the duration of the stop process, plus the SSD on a stop trial is smaller than the go process RT, then the go action can be successfully inhibited (successful stop). The interaction of the basic go and stop processes can be influenced by the dynamics of the proactive delayed responding as well as the dynamics of the attentional modulation of reactive stopping. The PRAD model enables obtaining the full posterior distributions of SSRT, proactive delay in responding of the go process, and the probability of proactive cognitive states at a trial level.

We defined the 3 latent dynamic measures examined in this study as follows. SSRT was the posterior mean of the duration of the stop process on correct or incorrect stop trials; it was not defined for go trials or trials on which a participant responded before the stop signal was shown. Probability of proactivity was the posterior mean of $\rho_t$, i.e., the posterior probability of $\rho_t = 1$. Proactive delaying was the posterior mean of $\lambda_t$ on trials with probability of proactivity greater than 0.5, otherwise 0. Probability of proactivity and proactive delaying were defined for all trials.

For further details of the model and inference, see SI Methods and ref. [50].

## Between- and within-subjects, general linear model analysis of fMRI

We fit general linear models to the fMRI BOLD recordings using Nilearn's FirstLevelModel. Condition and parametric regressors were modeled as impulses, with a duration of 0, and convolved with the SPM software's double gamma hemodynamic response function and the function's time derivative. Before fitting, the BOLD signal was scaled to percent signal-change from the mean in the time dimension. An AR1 model was used to whiten the data and design matrices to account for temporal autocorrelation in the BOLD signal.

To determine between-subjects associations between subject-average brain activity and behavioral measures, for each subject and voxel, we fit the model

$$BOLD(t) = \beta_0 + (HRF*Conditions)(t) + Nuisance\,(t) + \epsilon(t) \quad \text{(Model 1)}$$

BOLD(t) is the BOLD signal of the voxel at time $t$ ($t \in \{1, \ldots, T\}$ for $T$ the total number of volumes acquired); (HRF*Conditions)(t) is the value at $t$ of the convolution with the hemodynamic response function HRF of condition regressor(s) Conditions; Nuisance($t$) is the effect at $t$ of nuisance regressors, which were 6 motion parameters (translational and rotational displacement along each of three axes) and 6 cosine basis functions (corresponding to high-pass filtering at 0.01 Hz); and $\epsilon(t)$ is the model's error at $t$. We fit models with three sets of condition regressors:

$$Conditions = \beta_1(I_{Go} + I_{Stop}). \quad \text{(Conditions 1)}$$

$$\text{Conditions} = \beta_1 I_{\text{Go}} + \beta_2 I_{\text{Stop}}. \qquad \text{(Conditions 2)}$$

$$\text{Conditions} = \beta_1 I_{\text{Correct go}} + \beta_2 I_{\text{Incorrect go}} + \beta_3 I_{\text{Correct late go}} + \beta_4 I_{\text{Incorrect late go}}$$

$$+ \beta_5 I_{\text{No response go}} + \beta_6 I_{\text{Correct stop}} + \beta_7 I_{\text{Incorrect stop}} + \beta_8 I_{\text{SSD stop}}$$
$$\text{(Conditions 3)}$$

$I_{\text{Condition}}$ is an indicator function indicating when the subject experiences Condition (for example, $I_{\text{Go}}$ is 0 except at the moment when a subject is presented with a go trial). For between-subjects analyses, we used Model 1 with Conditions 1 to obtain task activation ($\beta_1$); used Model 1 with Conditions 2 to obtain go activation ($\beta_1$) and stop activation ($\beta_2$); and used Model 1 with Conditions 3 to obtain correct stop versus correct go activation ($\beta_6 - \beta_1$), incorrect stop versus correct go activation ($\beta_7 - \beta_1$), and incorrect stop versus correct stop activation ($\beta_7 - \beta_6$). For correlations between these activations and subject-average behavioral measures, we computed each subject-average measure as the mean of the measure over the trials from both runs during which it assumed values; that is, SSRT was averaged over correct and incorrect stop trials, probability of proactivity and proactive delaying over all trials, and go RT over go trials with a recorded response.

To determine within-subjects associations between trial-level brain activity and behavioral measures, for each subject and voxel, we fit the model

$$\text{BOLD}(t) = \beta_0 + (\text{HRF*Conditions})(t) + (\text{HRF*Modulation})(t)$$
$$+ \text{Nuisance}(t) + \epsilon(t). \qquad \text{(Model 2)}$$

(HRF*Modulation)($t$) is the value at $t$ of the convolution with the hemodynamic response function of the parametric regressor Modulation. To investigate within-subjects associations between brain activity and SSRT on stop trials, probability of proactivity on all trials, proactive delaying on all trials, and observed RT on go trials, we set Modulation $= \beta_3$ SSRT, Modulation $= \beta_3$ P(Proactive), Modulation $= \beta_3$ Proactive delaying, and Modulation $= \beta_3$ Go RT, respectively, and used Model 2 with Conditions 2. Each of SSRT, P(Proactive), Proactive delaying, and GoRT was standardized over the conditions during which it assumed values by subtracting its mean and dividing by its standard deviation; that is, SSRT had mean 0 and standard deviation 1 over correct and incorrect stop trials (and was 0 on all other trials), P(Proactive) and Proactive delaying had mean 0 and standard deviation 1 over all trials, and GoRT had mean 0 and standard deviation 1 over go trials with a recorded response (and was 0 on all other trials).

We fit these regression models for each subject and each of their 2 SST runs. For each model, subject, and voxel, we combined the regression results from the 2 runs with a fixed effects model through FirstLevelModel's compute_contrast method. For each model and voxel, we estimated the effect of each scanner as the mean of the regression coefficients of the subjects who were scanned by it minus the grand mean of the regression coefficients of all subjects. Then, we adjusted the regression coefficients by subtracting the estimated scanner effects. All analyses of the regression coefficients used these adjusted values. We used a 1-sample Cohen's $d$ (sample mean divided by sample standard deviation) to measure the effect sizes of regression coefficients. Whole-brain correlations and Cohen's $d$ values were visualized on the cortical surface[75]. The surface was obtained using the fetch_surf_fsaverage function in Nilearn's struct module (mesh = "fsaverage"); data were sampled to the surface using the vol_to_surf function in Nilearn's surface module; and plotted using the plot_surf_stat_map function in the surf_plotting module.

## Networks and regions of interest

We extracted the whole-brain regression coefficients in 2 networks and 3 sets of regions. We used the Shirer networks for our primary analyses and used the Yeo−17 networks, Shirer regions, cognitive-control regions, and subcortical regions to test the stability of our within-subjects findings. For each subject and each regression coefficient of interest, we obtained the coefficient's value in each area (network or region) by calculating the mean of the subject's coefficients over the voxels belonging to the area. We used these area-average regression coefficients of each subject: to compute Cohen's $d$ and Pearson $r$ values for the network-level comparison of between- and within-subjects associations (Figs. 3 and 5) and the measurement of Simpson's paradox (SI Fig. S3); to compute Cohen's $d$ values for the subgroup (Fig. 7), stability (Fig. 8), and task run robustness (SI Fig. S7) analyses; and to view distributions over subjects of the regression coefficients (SI Fig. S2).

The Shirer networks and regions were obtained from ref. 54. To obtain the voxel-coordinates of the Yeo-17 networks, we used a mapping between the Brainnetome[76] and Yeo atlases[77]. We assembled the cognitive-control regions to include 8 areas activated by the SST, 2 core default mode areas, and 1 core salience network and error-processing area. The areas activated by the SST were taken from a meta-analysis of 70 inhibitory control studies[30] (right anterior insula, right caudate, right inferior frontal gyrus, right middle frontal gyrus, right presupplementary motor area, and right supramarginal gyrus) and a study that segmented high-resolution structural MRI[78] (left and right subthalamic nucleus). To obtain the default mode areas (posterior cingulate cortex and ventromedial prefrontal cortex), we retrieved a Neurosynth automatic meta-analysis of 777 studies for the term "default mode" on 2024-02-05; extracted clusters from this map using the connected_regions function in Nilearn's region_extractor module with keyword argument "extract_type" set to "connected_components"; identified by eye the clusters corresponding to the posterior cingulate and ventromedial prefrontal cortex; and for each cluster defined the region to be the 6 mm cube centered on the voxel with the highest meta-analysis Z-score in the cluster. To obtain the salience network and error-processing area (dorsal anterior cingulate cortex), we retrieved a Neurosynth automatic meta-analysis of 464 studies for the term "error" on 2023-09-21 and defined the region to be the 6 mm cube centered on the voxel with the highest meta-analysis Z-score. We obtained subcortical regions from a subcortical probabilistic atlas[79]. We resampled the atlas's probabilistic subcortical labels in 1 mm cubed MNI152 2009c nonlinear asymmetric space to the 2 mm cubed MNI152 space of our SPM echo-planar imaging template using resample_to_img from Nilearn's resampling module, and then thresholded these probabilistic maps at 0.5 to obtain region masks.

## Stability analysis

To assess the stability of the within-subjects results, regression coefficients were resampled at varying sample sizes and the correlation was evaluated against the results in the full sample. Specifically, for each of SSRT, probability of proactivity, and proactive delaying, in each set of networks or regions: 10,000 samples of $n$ subjects were drawn with replacement; the Cohen's $d$'s of each sample's regression coefficients were calculated and Pearson correlated with the Cohen's $d$'s of the full sample over the regions or networks; and the mean and 95% and 99% bootstrap confidence intervals were calculated of the correlation (SSRT and probability of proactivity $n = 25, 40, 70, 120, 200, 335, 560, 935, 1570, 2635, 4423$; proactive delaying $n = 25, 40, 70, 115, 195, 320, 535, 895, 1490, 2480, 4137$). The 95% and 99% bootstrap confidence intervals were calculated, respectively, as the intervals covering the 2.5th to 97.5th percentiles and 0.5th to 99.5th percentiles of the 10,000 correlations at each $n$.

We also directly examined the distributions of the Cohen's $d$'s of the resamples as a function of $n$ in regions of interest. Specifically, for each of SSRT, probability of proactivity, and proactive delaying, in

each region of interest: 10,000 samples of $n$ subjects were drawn with replacement; the Cohen's $d$ of each sample's regression coefficients was calculated; and the mean and 95% and 99% bootstrap confidence intervals were calculated of the Cohen's $d$ (SSRT and probability of proactivity $n = $ 25, 40, 70, 120, 200, 335, 560, 935, 1570, 2635, 4423; proactive delaying $n = $ 25, 40, 70, 115, 195, 320, 535, 895, 1490, 2480, 4137). The 95% and 99% bootstrap confidence intervals were calculated as the intervals covering, respectively, the 2.5th to 97.5th percentiles and 0.5th to 99.5th percentiles of the 10,000 Cohen's $d$'s at each $n$.

### Representational similarity analysis

For each Shirer network, and for each subject, we computed the correlation between the subject's within-subjects brain maps of SSRT and probability of proactivity, SSRT and proactive delaying, and probability of proactivity and proactive delaying over the voxels in the network. Density estimates used Seaborn's kdeplot function with each distribution of correlations over subjects normalized to 1 (common_norm = False) and limited to values between −1 and 1 (clip = [−1,1]); all other parameters, including those determining the kernel smoothing bandwidth, were kept at their defaults.

### Statistical testing

Permutation tests were used for all statistical testing. The tests used two-sided alternatives and 10,000 resamples and were performed with Scipy's permutation_test function. FDR correction was performed using the Benjamini-Hochberg procedure with Scipy's false_discovery_control function. $P_{FDR}$ denotes an FDR-corrected $P$ value.

For the Shirer networks and an fMRI regression, we tested the null hypothesis for each network that the regression coefficients in the network had a mean of 0 (Figs. 3 and 5) by computing the means of resamples in which the signs of the coefficients were randomly chosen (permutation_test permutation_type = "samples"). Then, FDR correction was applied to the $P$ values of all networks (e.g., FDR correction was applied to the $P$'s of SSRT's regression coefficients over the Shirer networks).

For the Shirer networks, an fMRI regression, and a behavioral measure, we tested the null hypothesis for each network that the Pearson correlation between subject-average regression coefficients in the network and subject-average behavioral measures was 0 (Figs. 3 and 5) by computing the Pearson correlations of resamples in which regression coefficients were randomly paired with behavioral measures (permutation_test permutation_type = "pairings"). Then, FDR correction was applied to the $P$ values of all networks (e.g., FDR correction was applied to the $P$'s of correlations between correct stop versus correct go activation and SSRT over the Shirer networks).

For the Shirer networks and a param $\gamma_1$ or $\theta_1$, we tested the null hypothesis for each network that the mutually exclusive subgroups of subjects with param < 0 and with param > 0 had different mean regression coefficients (Fig. 7) by computing the differences between the means for resamples in which coefficients were randomly assigned to param < 0 and param > 0 (permutation_test permutation_type = "independent"). Then, FDR correction was applied to the $P$ values of all networks (e.g., FDR correction was applied to the $P$'s of mean differences between $\gamma_1 < 0$ and $\gamma_1 > 0$ over the Shirer networks).

For the Shirer networks, we tested the null hypothesis for each network that there was no difference in the network between the median correlation of SSRT and probability of proactivity and the median correlation of SSRT and proactive delaying; the median correlation of SSRT and probability of proactivity and the median correlation of probability of proactivity and proactive delaying; and the median correlation of SSRT and proactive delaying and the median correlation of probability of proactivity and proactive delaying (Fig. 6). We computed the differences between the medians of resamples in which correlations were randomly exchanged within subjects (permutation_test permutation_type = "samples"). Then, FDR correction

was applied to the $P$ values of all comparisons in all networks. (Since there are 14 Shirer networks and 3 tests per network, FDR correction was applied over 14 × 3 $P$'s.)

### Software

Data were processed and analyzed using Python (version 3.9.16), Scipy (version 1.11.4), Seaborn (version 0.13.2), Nilearn (version 0.10.1), FSL FLIRT (version 6.0), MATLAB (version R2020b), PyMARE (version 0.0.10), and JAGS[73] (version 4.3.0). Brain maps used the vik colormap[80].

### Reporting summary

Further information on research design is available in the Nature Portfolio Reporting Summary linked to this article.

## Data availability

Data used in this study were from the ABCD study (https://abcdstudy.org/), held in the National Institute of Mental Health Data Archive. These data are available to eligible researchers. Source data are provided with this paper.

## Code availability

All code used in this study has been archived at https://doi.org/10.5281/zenodo.18626601.

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

## Acknowledgements

Data used in the preparation of this article were obtained from the Adolescent Brain Cognitive Development℠ (ABCD) Study (https://abcdstudy.org), held in the NIMH Data Archive. This is a multisite, longitudinal study designed to recruit more than 10,000 children age 9–10 and follow them over 10 years into early adulthood. The ABCD Study® is supported by the National Institutes of Health and additional federal partners under award numbers U01DA041048, U01DA050989, U01DA051016, U01DA041022, U01DA051018, U01DA051037, U01DA050987, U01DA041174, U01DA041106, U01DA041117, U01DA041028, U01DA041134, U01DA050988, U01DA051039, U01DA041156, U01DA041025, U01DA041120, U01DA051038, U01DA041148, U01DA041093, U01DA041089, U24DA041123, U24DA041147. A full list of supporters is available at https://abcdstudy.org/federal-partners.html. A listing of participating sites and a complete listing of the study investigators can be found at https://abcdstudy.org/consortium_members/. ABCD consortium investigators designed and implemented the study and/or provided data, but did not necessarily participate in the analysis or writing of this report. This manuscript reflects the views of the authors and may not reflect the opinions or views of the NIH or ABCD consortium investigators. The ABCD data repository grows and changes over time. The ABCD data used in this report came from https://doi.org/10.15154/1523041. Some of the computing for this project was performed on the Sherlock cluster. We would like to thank Stanford University and Stanford Research Computing for providing computational resources and support that contributed to these research results. This work was supported by National Institutes of Health MH121069 (V.M.), MH124816 (W.C.), National Science Foundation No. 2024856 (V.M.), and the Stanford Maternal and Child Health Research Institute (W.C., P.M.).

## Author contributions

Study design: P.K.M., N.K.B., Z.G., W.C., V.M.; Analysis: P.K.M., N.K.B., Z.G.; Interpreting results: P.K.M., N.K.B., Z.G., W.C., V.M.; Writing: P.K.M., N.K.B., V.M.; Editing: P.K.M., N.K.B., Z.G., W.C., V.M.; Supervision: P.K.M., V.M.

## Competing interests

The authors declare no competing interests.
