## [Transparent Peer Review file · Nature Communications]

Nonergodicity and Simpson's paradox in neurocognitive dynamics of cognitive control

Corresponding Author: Mr Nicholas Branigan

Version 0:

Reviewer comments:

Reviewer #1

(Remarks to the Author)
Nat Comm review

Nonergodicity and Simpson's paradox in neurocognitive dynamics of cognitive control
Mistry et al.,

In this manuscript, the authors tease apart between and within-subject neural dynamics focusing on ergodicity. It makes several interesting and important claims, and the most central claim, about non-ergodicity, is highly likely 'true'. At the same time, there are multiple problems with the manuscript, which we attempt to group below.

One challenge is the use of ergodicity in various sections. At present, it is rather bit imprecise, e.g. 'Nonergodicity refers to a system's behavior where time averages and ensemble averages do not coincide' or 'specifically that the average trend in the group reflects the trend within each individual', neither of which is a sufficiently precise definition. The manuscript cites, but doesn't really engage with, the literature on this topic. Ergodicity implies that sampling one individual repeatedly will yield the same sampling characteristics as multiple individuals once. (see, for instance, Molenaar & Campbell, 'Condition 1: Homogeneity The first condition for ergodicity is that each subject in the population has to obey the same statistical model (homogeneity of the population). This means that the main features of a statistical model describing the data are invariant across subject'

Take, for instance, IQ. Ergodicity for IQ would imply every individual's mean IQ is 100, and standard deviation is 15. This is even stricter than demanding that, say, the associations between brain and behaviour are the same - It means that there are no stable individual differences.

'While progress has been made in understanding nonergodicity within behavioral contexts its application to human brain function remains unexplored' this is a bit too strong - almost a decade ago we wrote this paper tackling exactly the same problem <https://www.biorxiv.org/content/10.1101/039412v1.abstract>. Admittedly, this remains a preprint (never published due to modest sample size) so it is far from the last word (and the authors should feel no obligation to cite it as we have a clear coi). There are also plenty of other papers discussing ergodicity and fMRI, more and less recent, e.g. Medaglia, J. D., Ramanathan, D. M., Venkatesan, U. M., & Hillary, F. G. (2011). The challenge of non-ergodicity in network neuroscience. *Network: Computation in Neural Systems*, 22(1-4), or Hunter, M. D., Fisher, Z. F., & Geier, C. F. (2024). What ergodicity means for you. *Developmental Cognitive Neuroscience*, 101406. So we don't quite understand why the paper claims this topic is 'unexplored'

-A related challenge is that the introduction is framed mostly as a question of generalizability, but the better way to look at it is as two fundamentally different, but both worthwhile, questions. It is not true that we only ever do between-subject work to generalize to within-subject processes - the finding that faster people are also more accurate is profound and useful even though it does not generalize to the intra-individual level. The problem is when people jump from one explanatory level (and accompanying question) to the other. The fault lies in the inference more than the design. So, the suggestion 'Analyzing data from a single individual over time can yield different conclusions than analyzing data across multiple individuals at a single point in time' or 'or even misleading conclusions' is misframed - if done well, both are legitimate approaches to different questions, so there is no intrinsic problem with different conclusions. The problem arises in the implicit generalisation, which we agree with the authors happens too often - yet this is the fault of the inferences, not the study designs.

PRAD model

-The paper leans heavily on the PRAD model, which at present is still a preprint. We are fans of preprints, yet everything follows from this model. The virtues of the PRAD models are extensively extolled, but the only validation is elsewhere. The lack of code availability compounds this issue, and our foray into the other paper did not tackle all questions we had

I don't know what 'However, in addition to typical drift-diffusion process parameters, PRAD includes additional individual trait-like and dynamic trial-level measures' means precisely - trait-like measures of what? Drift diffusion models include traitlike measures. The paper leans hard into the virtues of the not yet published prad model, but doesn't really contextualize it amongst alternatives.

- There is very little discussion, and robustness tests on how/or if certain modeling assumptions could lead to the diverging results you find between parameters. Moreover, the behavioural covariances between parameters are not presented (as far as we can find).

It is unclear whether ABCD is the right dataset for this particular question - it has a very large N, but a relatively modest T (it is an example of the denseness of the paper that it is very hard to find the actual number of trials - the figures suggest 60, which is not very much at all for SSRT), which surely affects power and inferences when comparing the two 'directions;' of inference. For the goals of this paper, we think an imaging study with a standard number of subjects (n ~ 50) yet a very dense design might be just as good or even better?

There is an overemphasis on the number of subjects with little discussion on the number of trials. Yet, the stability of your parameters is based on T. There needs to be split-half reliability, first for model parameters, and then for neural stability. This could be done by correlating parameter estimates with e.g. subsets of trials, or across ABCD waves. There are two reasons why reliability is particularly crucial for this paper; 1) necessarily prerequisite when developing a novel computational model and 2) inter/intra individual differences are fundamental to the conclusions of the paper. That being said, when done robustly it could go in the SI – with a single sentence sufficing.

Moreover, we know this fMRI paradigm has shockingly horrible reliability (Kennedy et al., 2022, Neuroimage);

"Mean reliability and stability in ROIs, averaged across the targeted contrasts for all 3 tasks in the full "QC" dataset, was .076 (SD=.060) for within-session reliability at baseline, .100 (.068) for within-session reliability at follow-up, and .072 (.066) for longitudinal stability (Figure 2A)"

Whereas this paper states

"The within-subjects results in samples as small as 25 subjects correlated strongly with the results in the full sample."

This may very well be a failure of our imagination, but it seems hard to reconcile these two findings

Density

The paper in the present form is almost unreadably dense - It tries to do so many things at once that the reader (or at least, both of us) gets lost. One could easily imagine any of the 5 aims to be a paper in and of itself. We found it at times very hard to follow even with our (relatively) similar backgrounds and interests, and in its current form it seems inaccessible to most readers. There are multiple atlases, a new model, multiple lines of thought, a somewhat surprising RSA analysis, all fleshed out to varying degrees. The amount of work is impressive, yet it does not succeed in turning it into a coherent, followable paper.

Minor points

This paper does not have an excessive number of acronyms, yet this needs to be considered in the model preprint paper. For a paper so dependent on modeling the code should be made available at this stage, not upon publication.

Figure 3 (and all neuroimaging figures) are using the Jet colormap. This is 1) non-perceptually uniform, 2) not grayscale friendly, or 3) colorblind friendly. See Crameri et al., 2020 Nat. Comm. on the issues regarding these color maps.

On the whole, this is a paper on an important topic with an impressive amount of work, but that ultimately fails to convince through the sheer disparity of analyses which necessitate overly condensed, sometimes superficial treatments, and leave open many of the obvious challenges. It is possible that some of the issues we raise were in fact treated in some way - but we can assure the authors that both of us made a genuine effort to (re)read the paper closely.

In the interest of openness, we sign our reviews
Rogier Kievit & Nick Judd

(Remarks on code availability)

The authors have not shared their code (they state they will do so upon publication, but I cannot see any compelling reason for this decision)

Reviewer #2

(Remarks to the Author)

This is a very interesting study on ergodicity by Mistry et al, which examined between-person (BP) and within-person (WP) brain-behavior associations using a computational model of the stop signal task in ABCD data. Mistry et al. found robust evidence of nonergodicity and Simpson's Paradox, such that the BP and WP relationships had different directions. The biggest strength of this study is its direct ability to examine WP processes. I will focus my review mostly on ergodicity as I believe my expertise is more useful there (the computational task modeling and fMRI analyses all appear excellent). The manuscript reads well, the study is analytically impressive, and the results have important implications for understanding of cognitive control processes at an individual-level. Moreover, the overall theme relates to a growing emphasis on precision imaging and ergodicity in cognitive neuroscience. However, there are fundamental misunderstandings of ergodicity throughout the manuscript, perhaps related to not including key references (see comment #1). These fundamental issues led to serious flaws in framing the issue of nonergodicity and how analyses were performed. Major issues include not recognizing what non-ergodicity means (all-or-nothing), leading to a focus on testing it rather than the importance of WP processes (comment #2), the authors making their own fallacy in assuming ergodicity when averaging WP results (comment #4), and conflating nonergodicity and Simpson's Paradox in their analyses (comment #9). There are additional major concerns related to framing and which analyses were chosen as main vs. robustness, described below. While these are major issues, they are quite understandable (and common mistakes) as ergodicity is a difficult concept to understand and is relatively new to cognitive neuroscience. I believe they are all addressable issues and this could be an important manuscript if addressed. I would look forward to reviewing a revised version of this manuscript.

Major Comments

Introduction

1. Lines 55-57: "Nonergodicity [...], its application to human brain function remains unexplored" is incorrect. While this is one of the few studies to empirically test nonergodicity in this field, there are several articles that have explicitly focused on ergodicity in brain functioning. Most of these articles were not cited in the manuscript despite being highly relevant. Beyond it being responsible to cite these papers as they are of utmost relevance, referencing them is also needed as key takeaways from these articles would aid some of the nuanced misunderstandings in the present manuscript (see other points below).

Relevant citations include, chronologically:

- a. Medaglia, J. D., Ramanathan, D. M., Venkatesan, U. M., & Hillary, F. G. (2011). The challenge of non-ergodicity in network neuroscience. *Network* (Bristol, England), 22(1–4), 148–153. <https://doi.org/10.3109/09638237.2011.639604>
- b. Kraus, B., Zinbarg, R., Braga, R. M., Nusslock, R., Mittal, V. A., & Gratton, C. (2023). Insights from personalized models of brain and behavior for identifying biomarkers in psychiatry. *Neuroscience & Biobehavioral Reviews*, 152, 105259. <https://doi.org/10.1016/j.neubiorev.2023.105259>
- c. Hunter, M. D., Fisher, Z. F., & Geier, C. F. (2024). What ergodicity means for you. *Developmental cognitive neuroscience*, 68, 101406. <https://doi.org/10.1016/j.dcn.2024.101406>
- d. Gell, M., Noble, S., Laumann, T. O., Nelson, S. M., & Tervo-Clemmens, B. (2024). Psychiatric neuroimaging designs for individualised, cohort, and population studies. *Neuropsychopharmacology*, 1–8. <https://doi.org/10.1038/s41386-024-01918-y>
- e. Mattoni, M., Fisher, A. J., Gates, K. M., Chein, J., & Olino, T. M. (2025). Group-to-individual generalizability and individual-level inferences in cognitive neuroscience. *Neuroscience & Biobehavioral Reviews*, 169, 106024. <https://doi.org/10.1016/j.neubiorev.2025.106024>

2. A nuanced, but critical, misunderstanding (in my view) throughout the introduction is the goal of testing nonergodicity. In my view, it is neither important nor interesting to explicitly test for nonergodicity in human processes. Molenaar (2003) established, both theoretically and through a statistical 'proof', that essentially every human process is inherently nonergodic. Critically, ergodicity is all-or-nothing, requiring homogeneity between persons and stationarity within persons. From Molenaar 2003 alone, we can confidently assume that cognitive control processes would be nonergodic. Thus, if the goal is simply testing nonergodicity, this manuscript does not provide much incremental knowledge. However, this is really just an issue of framing. Testing the within-person process itself (rarely studied) is interesting and important, and the results here are indeed a striking and useful demonstration for the field to recognize that between-person processes are distinct and independent from within-person processes (an issue of group-to-individual generalizability, slightly different from ergodicity, see Fisher 2018). Unless the authors have a strong argument otherwise, I believe much of the introduction needs reframing based on this nuance, perhaps more toward an emphasis on why within-person study is important (and distinct from between-person study). Several of the references in comment #1 could aid this reframing.

3. Lines 112-118: "[...] this conventional approach implicitly assumes ergodicity [...]" While there are relevant ergodic assumptions (which the authors overlook themselves in the present analyses, see comment 4), the intended point here is not necessarily true. The assumption of ergodicity would only be relevant if the between-person study was used for within-person conclusions (again, the within-person aspect is the key here and not sufficiently addressed in the introduction). If researchers use between-person SST tests to make between-person conclusions, nonergodicity is not highly relevant as the authors describe here.

Main

4. The authors make their own ergodic assumption fallacy in their tests of within-person processes. From my understanding, they estimated within-person relationships entirely ideographically and then aggregated them into a single within-person effect size. Importantly, ergodicity is not the equivalence of the average within-person relationship with the between-person relationship, but the equivalence of literally every single person's within-person relationship with the between-person relationship (hence, why we can safely assume human processes are non-ergodic without explicitly testing them). This point is made clear in Molenaar 2003, Fisher 2018, Hunter 2024, and Mattoni 2025, particularly with the emphasis of the homogeneity assumption for ergodicity. At the minimum, it would likely be more appropriate to instead present a distribution of the individual's within-person estimates. If the authors are truly focused on a single aggregated value, alternative approaches such as the two-stage random effects meta-analysis presented by Lee and Gates 2024 (<https://doi.org/10.1080/00273171.2023.2229310>) would be more appropriate than the approach used here.

5. The authors were focused on the sign of the relationship (i.e., positive vs. negative) throughout the manuscript. Another striking finding is that the within-person effect sizes were substantially higher. Can the authors speculate as to why this may be, and potentially address this in the manuscript? Importantly, ergodicity is not simply the same direction of the relationship, but literal equivalence.
 6. Figure 5 would be improved by marking the point at which sample size = number of trials (I could not find the number of trials in manuscript). As Fisher 2018 pointed out, there are important power implications for differing observations for between-person (n) and within-person (t) study. This should be explicitly addressed in the manuscript.
 7. The authors provide a useful robustness check in the supplement focusing between-person SSRT study on stop trials and proactivity stud on all trials, which they state “aligns more closely with our within-subjects analyses.” While the results are mostly consistent with the main results presented, they are not as striking and it is unclear why this approach is a robustness check instead of the main analysis, when the goal of the present paper is to compare BP and WP relationships.
 8. Relatedly, supplementary Figure S2 panel C, this between-person result seems to much more closely resemble the within-person results in Figure 2. This is not consistent with the author’s description of robustness checks in the main manuscript. This needs explicit clarification.
 9. Aim 5, “Brain networks exhibit varying degrees of nonergodicity and a hierarchical organization by nonergodicity” is flawed; the authors erroneously conflate Simpson’s Paradox and nonergodicity. Again, nonergodicity is all-or-nothing and ergodic assumptions are violated with ANY non-equivalence, not necessarily a difference in relationship direction (Simpson’s Paradox). If the author’s maintain the present analysis, this should be renamed to measuring Simpson’s paradox. That said, I’m not sure why quantifying where Simpson’s paradox occurs more often is informative. Instead, it may be more useful to understand where in the brain WP relationships are +, WP relationships are -, BP relationships are +, BP relationships are -. The point of ergodicity is that BP and WP relationships should be considered separately.
- Discussion
10. Much of the discussion would need reframing based on the context I described in comment #2.
 11. Overall, the strength of this study is its unique ability to examine WP brain-behavior relationships. More time should be spent explaining why that is important, what the authors learned that BP study cannot tell us, and implications of those WP relationships.
 12. Related to the importance of WP study, perhaps the authors could speculate on important directions forward for that goal, whether in terms of study design or computational approaches.

Minor

1. Figure 1 panels A-C are very useful. More attention could perhaps be spent on Panel D, which did not provide me much clarity on the study goals without several looks and reads of the caption.
2. Lines 67-70: “Group-level relationships may obscure the dynamic processes occurring within each individual [...]” More important nuance here. It is not so much that group-level relationships obscure individual-level relationships, but that they are entirely distinct and the study of between-person relationships cannot be used to inform individual-level relationships without ergodic assumptions.
3. Lines 74-76: “[...] paving a way for personalized interventions [...] there is a pressing need to integrate [...] in inhibitory control.” While I agree with the notion, this statement was not well set up by the introduction, which spent little time on why nonergodicity is important for the study of within-person processes, which I believe is what the authors are referencing here.
4. I find the section “Adaptive regulation of inhibitory control associated with distinct within-subjects results over subgroups” very interesting; it relates individual differences in the WP brain-behavior processes (rather than averaging them) to real behavioral differences. This was a joy to see and may be worth emphasizing more in the manuscript.
5. I’m not sure what “a paradigm shift [...] toward nonergodicity neuroscience” (line 671) means. That read a bit grandiose.

(Remarks on code availability)

The authors provided example code for the computational model they used for the SST. This code appropriately describes their modeling. However, it is not specific to the ABCD data and there is no code for the fMRI analysis. Since ABCD is public data, this code would be a useful resource for the field if the authors are willing to provide it. As currently provided, the code cannot be used to reproduce the results.

Version 1:

Reviewer comments:

Reviewer #2

(Remarks to the Author)

The revised manuscript by Mistry, Brannigan et al. uses a computational modeling task in the ABCD study to examine distinct between-person and within-person brain-behavior relationships in a cognitive control task. I maintain my perspective from the initial version that the topic is important and timely and that the methods are sound to test the research question. In this revision, the authors appropriately addressed my initial major concerns about the framing of ergodicity throughout the manuscript as well as some methodological refinements. I also wish to note I found this version of the manuscript much easier to digest and more impactful than the initial version, which I appreciate reviewer 1’s comment for prompting and the authors efforts for addressing. Overall, I believe the authors addressed all my concerns and that this manuscript would be a strong contribution to growing interest in ergodicity in psychology and neuroscience.

(Remarks on code availability)

Response to Reviewers

Reviewer #1

[Comment 1.1] *In this manuscript, the authors tease apart between and within-subject neural dynamics focusing on ergodicity. It makes several interesting and important claims, and the most central claim, about non-ergodicity, is highly likely 'true'.*

[Response 1.1] We thank the reviewers for their positive comments.

[Comment 1.2] *At the same time, there are multiple problems withto the manuscript, which we attempt to group below.*

One challenge is the use of ergodicity in various sections. At present, it is rather bit imprecise, e.g. 'Nonergodicity refers to a system's behavior where time averages and ensemble averages do not coincide' or 'specifically that the average trend in the group reflects the trend within each individual', neither of which is a sufficiently precise definition. The manuscript cites, but doesn't really engage with, the literature on this topic. Ergodicity implies that sampling one individual repeatedly will yield the same sampling characteristics as multiple individuals once. (see, for instance, Molenaar & Campbell, 'Condition 1: Homogeneity The first condition for ergodicity is that each subject in the population has to obey the same statistical model (homogeneity of the population). This means that the main features of a statistical model describing the data are invariant across subject. Take, for instance, IQ. Ergodicity for IQ would imply every individual's mean IQ is 100, and standard deviation is 15. This is even stricter than demanding that, say, the associations between brain and behaviour are the same - It means that there are no stable individual differences.

[Response 1.2] We thank the reviewers for this important comment, which has helped us clarify and sharpen the focus of our manuscript. In response to this comment, as well as related suggestions from Reviewer 2, we have substantially revised the Introduction and Discussion to more precisely articulate our framework and contributions.

The revised Introduction now includes a more precise definition of ergodicity and engages with the literature in greater depth. Please see multiple changes to the Introduction lines 40-85.

Additionally, in the Discussion, we now frame our findings in terms of what within-subjects analyses reveal about individual-level mechanisms rather than focusing narrowly on tests of ergodicity assumptions. We believe these revisions better position our work as addressing a critical gap in cognitive neuroscience: understanding whether and how brain-behavior mechanisms at the individual level differ from patterns observed at the group level, a question with profound implications for both basic research and clinical translation.

Per the Reviewer's suggestion our focus is not showing that cognitive processes fail to meet the strict definition of ergodicity, which as Reviewer 2 notes can safely be assumed, but rather to ask whether this violation of ergodicity is scientifically important. Our question is: Do brain-behavior relationships inferred from between-subjects analyses accurately represent the

47 mechanisms operating within individuals? This question is central to cognitive neuroscience,
where most studies examine between-subjects associations, but the ultimate goal also includes
obtaining a nuanced understanding of within-individual mechanisms.

Specifically, we examine the extent to which within-individual inferences diverge from, and can
even reverse, between-individual inferences about brain-behavior relationships. Our manuscript
demonstrates that ergodicity is not merely violated but also that this violation is scientifically
significant, fundamentally challenging interpretations of neural mechanisms that may be derived
solely from between-subject analyses, and underscoring the importance of within-subject
analyses.

Importantly, this question has not been systematically addressed in cognitive neuroscience from
a brain-behavior perspective, in part due to methodological limitations. Traditional experimental
paradigms and cognitive models have been unable to measure dynamic within-individual
variability at sufficient temporal resolution. Our novel contribution is leveraging dynamic
computational modeling to infer trial-level cognitive processes, enabling direct comparison of
brain-behavior associations at between-subjects and within-subjects levels.

We have revised the manuscript to clarify that our focus is on:

- The substantive differences between between-subjects and within-subjects brain-behavior
inferences
- The novel mechanistic insights revealed by within-subjects analysis that are unavailable
from conventional between-subjects approaches
- The practical and theoretical implications of these divergences for understanding
cognitive control

*[Comment 1.3] 'While progress has been made in understanding nonergodicity within
behavioral contexts its application to human brain function remains unexplored' this is a bit too
strong - almost a decade ago we wrote this paper tackling exactly the same problem
<https://www.biorxiv.org/content/10.1101/039412v1.abstract>. Admittedly, this remains a preprint
(never published due to modest sample size) so it is far from the last word (and the authors
should feel no obligation to cite it as we have a clear coi). There are also plenty of other papers
discussing ergodicity and fMRI, more and less recent, e.g. Medaglia, J. D., Ramanathan, D. M.,
Venkatesan, U. M., & Hillary, F. G. (2011). The challenge of non-ergodicity in network
neuroscience. *Network: Computation in Neural Systems*, 22(1-4), or Hunter, M. D., Fisher, Z. F.,
& Geier, C. F. (2024). What ergodicity means for you. *Developmental Cognitive Neuroscience*,
101406. So we don't quite understand why the paper claims this topic is 'unexplored'.*

**[Response 1.3]** We thank the reviewers for pointing us to these highly relevant studies, and we
have now cited them all. We do wish to note that, apart from the preprint, most of the papers
cited above are based on theory, thought experiments, or simulations, with limited empirical
evidence. Accordingly, we have changed our wording in the introduction from 'unexplored' to
the following:

“While progress has been made in understanding nonergodicity in behavior¹⁻⁶ and
establishing strong theoretical frameworks⁷⁻¹⁰, few studies have empirically addressed
nonergodicity in human brain function^{9,11,12}.” (Introduction lines 72-75)

**[Comment 1.4]** *A related challenge is that the introduction is framed mostly as a question of*
*generalizability, but the better way to look at it is as two fundamentally different, but both*
*worthwhile, questions. It is not true that we only ever do between-subject work to generalize to*
*within-subject processes - the finding that faster people are also more accurate is profound and*
*useful even though it does not generalize to the intra-individual level. The problem is when*
*people jump from one explanatory level (and accompanying question) to the other. The fault lies*
*in the inference more than the design. So, the suggestion ‘Analyzing data from a single*
*individual over time can yield different conclusions than analyzing data across multiple*
*individuals at a single point in time^{5,6,9}’ or ‘or even misleading conclusions’ is misframed - if*
*done well, both are legitimate approaches to different questions, so there is no intrinsic problem*
*with different conclusions. The problem arises in the implicit generalisation, which we agree*
*with the authors happens too often - yet this is the fault of the inferences, not the study designs.*

**[Response 1.4]** We thank the reviewer for this insightful comment, which has helped us reframe
our argument more precisely. We agree that between-subjects and within-subjects analyses
address fundamentally different scientific questions, and that each kind of question may be
valuable. In response to this comment, we have substantially revised the Introduction to clarify
that between-subjects and within-subjects analyses answer different questions, both of which
may be important. For example:

“This striking divergence between group-level and individual-level patterns illustrates
that between- and within-individuals analyses answer fundamentally different scientific
questions¹³. This example also shows that both between- and within-individuals questions
can be of scientific interest. However, many important questions in the study of the brain
and behavior are posed at the within-individuals level⁶, and answering these questions
requires either within-individuals analyses, or the strong assumption of ergodicity^{1,4}.”
(Introduction lines 53-59)

Our manuscript is not a critique of between-subjects research, but rather a demonstration that: (1)
between-subjects and within-subjects analyses can yield substantively different insights about
brain-behavior relationships in cognitive control, and (2) within-subjects analyses, enabled by
dynamic computational modeling, provide perspectives that are fundamentally inaccessible
through between-subjects approaches alone. Both levels of analysis are essential for a complete
understanding of cognitive neuroscience.

**[Comment 1.5]** *PRAD model*

*The paper leans heavily on the PRAD model, which at present is still a preprint. We are fans of*
*preprints, yet everything follows from this model. The virtues of the PRAD models are*
*extensively extolled, but the only validation is elsewhere. The lack of code availability*
*compounds this issue, and our foray into the other paper did not tackle all questions we had. I*
*don’t know what ‘However, in addition to typical drift-diffusion process parameters, PRAD*

*includes additional individual trait-like and dynamic trial-level measures' means precisely -*
 *trait-like measures of what? Drift diffusion models include traitlike measures. The paper leans*
 *hard into the virtues of the not yet published prad model, but doesn't really contextualize it*
 *amongst alternatives.*

*There is very little discussion, and robustness tests on how/or if certain modeling assumptions*
 *could lead to the diverging results you find between parameters.*

**Figure 3. Nonergodicity is exhibited by a directly observed behavioral measure. a,** Between-
 subjects analysis. Whole-brain correlation map of associations between subject-average go
 reaction time and brain activation (correct stop versus correct go activation). Thresholded at
 Pearson $r \geq 0.05$. **b,** Within-subjects analysis. Whole-brain Cohen's d map of associations
 between trial-by-trial go reaction time and brain activity. Thresholded at Cohen's $d \geq 0.1$. **c,**
 Network-level comparison. Between- and within-subjects associations of go reaction time and
 brain activity were visualized in brain areas. Statistical significance is indicated by colored
 asterisks: red for between subjects ($P_{FDR} < 0.01$) and blue for within subjects ($P_{FDR} < 0.01$).
 For both between- and within-subjects analyses: $N = 4423$.

**[Response 1.5]** We appreciate the reviewer's insightful comments. In response, we have
 performed extensive supplementary analyses, providing both an empirical (non-model-based)
 demonstration of nonergodicity and comprehensive robustness checks on PRAD testing a range
 of its modeling assumptions. Our inferred model-based within-subjects relationships were robust
 to changing various PRAD model assumptions and using control models. Moreover, we show
 striking nonergodicity using an observed behavioral measure, Go reaction time. Thus, our
 within-subjects relationships were not sensitive to the structure of PRAD and our observations of
 nonergodicity did not depend on a specific model of the task. Details:

 **i. Non-model-based demonstration of nonergodicity**

We now present in the main text the results of between- and within-subjects analysis of the
 observed reaction time on go trials (please see Results lines 233-254 and Figure 3; previously,
 these results were in the SI). This provides a non-model-based demonstration of nonergodicity,
 fully independent of the PRAD model. These results show that our finding of pervasive
 nonergodicity does not depend on the use of PRAD.

 **ii. Model code**

We have now shared all the code relevant to the PRAD model and supporting functions that can
be used to run the PRAD model (using JAGS code) on an input file compatible with the ABCD
raw .csv data files. A synthetic file with synthetic data has been provided to enable testing the
model. Moreover, all code for analyses in this manuscript on the model's outputs has now been
shared. This includes the code for making subject lists, preprocessing fMRI data, performing
fMRI general linear model analysis, aggregating and statistically testing results, and plotting
results. The computational model code can be found at
https://github.com/scsnl/2024_Mistry_PRAD/tree/main/PRAD_Behavioral_Dynamics and all
other code used in the study can be found at
https://github.com/scsnl/2025_nonergodicity_mistry_branigan.

**iii. Trait-like measures clarification**

The PRAD model's description has been changed to add exactly what trait-like (persistence in
belief updating, proclivity for proactive control strategies, degree of performance-monitoring-
based modulation, and sensitivity to attentional modulation) and dynamic trial-level measures
(trial level modulations of the probability of proactivity, proactive delaying, and SSRT) are
inferred by the PRAD model, in addition to the standard trait-like measures from standard drift
diffusion models (please see changes to Methods lines 747-751, section on Bayesian modeling of
cognitive dynamics). We also emphasize that for this study, the dynamic trial-level measures are
of interest.

 **Supplementary Figure S6. Within-subjects associations were robust to PRAD modeling**
 **assumptions. a-d,** Within-subjects Cohen's *d* maps for SSRT inferred by PRAD and 3 control
 models: the PRAD model (a), PRAD fit independently to each SST run (PRAD-IR) (b), a
 version of PRAD that does not account for drift rate variability on go trials (PRAD-G) (c), and a
 model that assumes variability in trial-level SSRT but without the attentional and proactive
 mechanisms (RVM) (d). **e-g,** Within-subjects Cohen's *d* maps for probability of proactivity
 inferred by PRAD and 2 control models: PRAD (e), PRAD-IR (f), and PRAD-G (g). **h-j,** Within-
 subjects Cohen's *d* maps for proactive delaying inferred by PRAD and 2 control models: PRAD
 (h), PRAD-IR (i), and PRAD-G (j). The Cohen's *d* maps were thresholded at Cohen's *d* ≥ 0.1 .
 PRAD SSRT and probability of proactivity $N = 4423$; PRAD proactive delaying $N = 4137$;
 PRAD-IR SSRT and probability of proactivity $N = 3478$; PRAD-IR proactive delaying $N =$
 3160; PRAD-G SSRT and probability of proactivity $N = 3913$; PRAD-G proactive delaying
 $N = 3619$; RVM SSRT $N = 3998$.

iv. Multiple robustness tests and comparison to control models

Results based on the PRAD model parameters are now supported by multiple sensitivity analyses (please see Results lines 414-451):

1. Sensitivity to changes in specific modeling assumptions (SI Figure S6 c,g,j).
2. Sensitivity to session-level (vs combined) application of the model (SI Figure S6 b,f,i).
3. Testing a Control model (SI Figure S6 d). The findings on SSRT related brain-behavior associations are robust to the application of a control model that estimates trial level SSRT without any of the additional components that PRAD incorporates. (That model is however unable to make inferences about the proactive components, thus establishing the need for the PRAD model.)

v. PRAD model validation

While our behavioral manuscript¹⁴ includes multiple aspects of validation, we have provided a high level summary (subset of the PRAD behavioral paper) below to showcase some aspects of this validation:

Key behavioral measures	Aggregated Means		Individual-level (observed vs PRAD)		
	Observed	PRAD	Correlation*	MSE improvement [^]	KL divergence improvement [^]
Mean Go RT (mS)	522	515	0.91	7%	29%
Mean Stop-failure RT (mS)	449	420	0.90	25%	34%
Mean Go - Stop RT (mS)	74	95	0.51	9%	70%
Mean post-Go RT (mS)	509	495	0.92	11%	36%
Mean post-Stop RT (mS)	544	527	0.86	19%	49%
Mean post-Go - post-Stop RT (mS)	-35	-32	0.45	23%	89%
Stop failure rate (SFR)	48%	50%	0.85	42%	52%
Go omission rate	7%	12%	0.90	29%	18%

Table R1. Summary of PRAD model validation measures. Aggregate measures show the mean values across all individuals. The individual-level correlation shows correlation between observed and PRAD values across individuals, MSE improvement shows the improvement (lower MSE) in mean squared error of each measure based on PRAD compared to a control model, KL divergence improvement shows improvement in KL divergence of each measure based on PRAD compared to a control model. * $p < 0.0001$. [^] improvement versus a control model that allows random variability in trial-level SSRT but does not incorporate proactive strategy or attentional modulation of stopping. Based on complete behavioral data ($N = 7787$).

[Comment 1.6] Moreover, the behavioural covariances between parameters are not presented (as far as we can find).

[Response 1.6] The key PRAD parameters relevant for this study are the dynamic trial level parameters—SSRT, probability of proactivity, and proactive delaying. We have provided the behavioral correlations for these parameters both within and between subjects below:

	Mean	Standard deviation	25th percentile	50th percentile	75th percentile
SSRT & Probability of proactivity	-0.032	0.093	-0.079	-0.023	0.02
SSRT & Proactive delaying	0.032	0.134	-0.053	0.04	0.13
Probability of proactivity & Proactive delaying	0.454	0.21	0.293	0.459	0.62

**Table R2. PRAD model parameter correlations within subjects.** For each pair of behavioral
measures, the Pearson r was computed between pairs of measures per subject across trials. For
each subject, the correlation was computed over trials that had values for both measures in the
pair. Trials for which measures have values: SSRT, stop trials; probability of proactivity, all
trials; proactive delaying, all trials. Subjects were excluded for whom the correlation could not
be computed between at least 1 of the 3 pairs (since one of the measures was constant) $N =$
4126.

	Pearson r	99% CI low	99% CI high
SSRT & Probability of proactivity	0.126	0.101	0.151
SSRT & Proactive delaying	-0.321	-0.349	-0.291
Probability of proactivity & Proactive delaying	0.514	0.483	0.545

**Table R3. PRAD model parameter correlations between subjects.** For each behavioral
measure, the mean value was computed per subject. For each pair of behavioral measures, the
Pearson r was computed between the pair of measures across subjects. Confidence intervals (CI)
were computed for the correlation using bootstrapping with 10,000 resamples and the bias-
corrected and accelerated method with SciPy. $N = 4423$.

*[Comment 1.7] It is unclear whether ABCD is the right dataset for this particular question - it*
*has a very large N , but a relatively modest T (it is an example of the denseness of the paper that*
*it is very hard to find the actual number of trials - the figures suggest 60, which is not very much*
*at all for SSRT), which surely affects power and inferences when comparing the two ‘directions;’*
*of inference. For the goals of this paper, we think an imaging study with a standard number of*
*subjects ($n \sim 50$) yet a very dense design might be just as good or even better? There is an*
*overemphasis on the number of subjects with little discussion on the number of trials. Yet, the*
*stability of your parameters is based on T .*

**[Response 1.7]** We have now clearly mentioned the number of trials, both in the Results and in
Figure 8. We would like to clarify that the number of trials for each subject is in fact 360 across
2 runs. 60 was only the number of stop trials, with another 300 go trials. Thus, we believe that
this is already a relatively dense design in terms of the experimental paradigm.

We thank the reviewers for drawing our attention to the importance of the number of time points
in within-subjects study, and we have now commented on this point in the manuscript:

“We note that while our study’s large number of subjects is important for assessing the
 stability and generalizability of our inferred within-subjects relationships, the power to
 detect a particular subject’s relationship depends not on the number of subjects but on the
 number of temporal observations per subject (in our case, the number of trials)¹. Thus,
 within-subjects research benefits from datasets that are large-scale in the number of
 subjects as well as large-scale in the number of time points collected.” (SI Supplementary
 Results lines 117-122)

 We think that a small number of subjects (n~50) would not be a good fit for our aims, since
 studies with smaller samples tend to have relatively more homogeneous populations (as they are
 typically collected from a single site, resulting in similar demographics). Since heterogeneity can
 drive nonergodicity, a small number of subjects may understate the extent of nonergodicity in the
 entire population. Thus, we think that our use of ABCD—a large, multi-site study with broad
 population representation—is a key strength of our work. The large sample size also allows us to
 conduct subsampling stability analysis, and capture specific subgroups that show differences in
 brain-behavior mechanisms. Studies with lower sample size may also overestimate effect sizes
 and tend to show low reproducibility^{15,16}.

 *[Comment 1.8] There needs to be split-half reliability, first for model parameters, and then for*
 *neural stability. This could be done by correlating parameter estimates with e.g. subsets of trials,*
 *or across ABCD waves. There are two reasons why reliability is particularly crucial for this*
 *paper; 1) necessarily prerequisite when developing a novel computational model and 2)*
 *inter/intra individual differences are fundamental to the conclusions of the paper. That being*
 *said, when done robustly it could go in the SI – with a single sentence sufficing.*

 **[Response 1.8]** We thank the reviewers for this suggestion. We examined reliability by
 comparing behavioral and brain measures between the first and second runs of the stop-signal
 task. (We chose not to randomly sample trials because trial-wise performance depends heavily
 on trial history, specifically anticipation- and feedback-driven proactive control¹⁷, so random
 sampling of trials would disrupt these sequential dependencies and distort model estimation.)
 Here, it is important to acknowledge that between-run consistency can be influenced by factors
 such as practice effects, fatigue, or strategic shifts (e.g., varying levels of proactive control from
 the first to the second run).

 **Behavioral reliability:** For behavioral measures, we have conducted split-half reliability
 analysis by examining between-run similarities in SSRT, probability of proactivity and proactive
 delaying (Table R4). We found that between-run similarities were high for proactive delaying
 (r=0.52) and moderate for SSRT (r=0.34) and probability of proactivity (r=0.26).

	Pearson r	99% CI low	99% CI high
SSRT	0.337	0.269	0.415
Probability of proactivity	0.26	0.206	0.316
Proactive delaying	0.515	0.474	0.552

 **Table R4. Model parameter split-half reliability.** For each behavioral measure, the mean value
 was computed per subject and run and the Pearson r was computed between run 1 and 2 across

subjects. Confidence intervals (CI) were computed for the correlation using bootstrapping with
10,000 resamples and the bias-corrected and accelerated method with SciPy. $N = 3478$.

Further, we note that these dynamic measures are explicitly designed to capture how cognitive
processes fluctuate across trials in response to task history, performance monitoring, and
strategic adjustments. Split-half reliability analysis, which assumes measurement of stable traits,
is not conceptually ideal to assess measures that are designed to vary systematically with context,
including from one run to the next. The estimation of these trial-level parameters is linked to
sequential patterns of behavior, and is expected to change across trials and runs. For example,
our model captures phenomena such as increasing stopping expectancy after consecutive go
trials and adjusting proactivity following stop-signal failures. These effects are, by design,
history-dependent and context-sensitive. This distinction is crucial: the key variables of focus in
this study are state-dependent dynamic processes.

**Brain-behavior reliability:** For brain measures, we have conducted split-half reliability
analyses of within-subject brain-behavior associations, first at the group level (Supplementary
Figure S7) and second at the individual level (Figure R1). At each of these levels, we have
extracted brain and behavioral measures independently for the first and second runs of the SST
and compared their similarity between the 2 runs. We found high similarity between results from
the first and second runs (SI Figure S7). For the within-subjects brain-behavior associations, the
Pearson correlations between group results in runs 1 and 2 were between 0.80 (SSRT) and 0.98
(Go RT). These results have been included in the Results (lines 440-443) and SI Figure S7.

**Supplementary Figure S7. Within-subjects associations were robustly observed between**
**task runs.** Group-average within-subjects brain-behavior associations were reliably observed
across the 2 SST runs. SSRT, probability of proactivity, and proactive delaying were obtained
from the PRAD model fit to each of the SST runs independently. Brain activity was regressed on
each behavioral measure, and associations were extracted in 2 brain parcellations (the Shirer
networks and regions). Dots denote brain areas and the shaded areas show 95% bootstrap
confidence intervals for the regression lines. Go RT $N = 4423$; SSRT and probability of
proactivity $N = 3478$, proactive delaying $N = 3160$.

**[Comment 1.9]** *Moreover, we know this fMRI paradigm has shockingly horrible reliability*
*(Kennedy et al., 2022, Neuroimage); “Mean reliability and stability in ROIs, averaged across*
*the targeted contrasts for all 3 tasks in the full “QC” dataset, was .076 (SD=.060) for within-*
*session reliability at baseline, .100 (.068) for within-session reliability at follow-up, and .072*

(.066) for longitudinal stability (Figure 2A)”. Whereas this paper states “The within-subjects
 results in samples as small as 25 subjects correlated strongly with the results in the full sample.”
 This may very well be a failure of our imagination, but it seems hard to reconcile these two
 findings.

 **[Response 1.9]** We thank the reviewers for this observation and the opportunity to clarify what
 may appear to be contradictory findings. In fact, these results are measuring different phenomena
 and are compatible. The Kennedy et al. (2022) findings refer to individual-level split-half
 reliability. Our statement about samples of N=25 refers to group-level stability of within-subjects
 effects, i.e., whether the group-average within-subjects brain-behavior associations observed in
 one random sample of 25 people resembles the group-average within-subjects associations
 observed in the full sample of ~4,000 individuals. This measures whether the group-average
 within-subjects effect is stable across different samples of individuals.

 **Figure R1. Neural split-half reliability.** The reliability of each subject’s brain measures was
 computed for the ABCD and Human Connectome Project studies. For each brain measure, the
 reliability was computed as the Pearson correlation between a subject’s run 1 and run 2 brain
 maps extracted in the Shirer networks. Half violin and box plots show the distribution of the
 reliability over subjects. Brain measures for the ABCD stop signal task: correct stop versus
 correct go, incorrect stop versus correct go, incorrect stop versus correct stop, go RT, SSRT,
 probability of proactivity, and proactive delaying. Brain measures for the Human Connectome
 Project: motor task, foot versus hand; gambling task, loss versus win; relational task, match
 versus relational processing; social task, random versus mental; emotion task, neutral versus fear;
 working memory task, 0-back versus 2-back. ABCD: $N = 4423$ for non-model-based brain
 measures (correct stop versus correct go, incorrect stop versus correct go, incorrect stop versus
 correct stop, Go RT); $N = 3478$ for SSRT and probability of proactivity; $N = 3160$ for
 proactive delaying. HCP: $N = 524$.

Further, to assess whether the data quality of the ABCD SST fMRI paradigm is unusually poor,
we conducted split-half reliability analysis on the 6 task fMRI paradigms from the Human
Connectome Project (HCP) (Figure R1). HCP median correlations were between -0.03
(gambling task) and 0.74 (emotion task). Both the ABCD and HCP studies showed wide
correlation distributions, indicating substantial variability in the reliability of individual subjects'
brain measures. While reliability values were somewhat higher for the HCP than the ABCD SST,
we attribute this not to lower data quality in the ABCD study but to a difference in task design.
All HCP fMRI tasks used a block design while the ABCD SST used an event related design.
Overall, we found that the reliability of brain measures was comparable between the ABCD SST
and the HCP.

*[Comment 1.10] Density: The paper in the present form is almost unreadably dense - It tries to*
*do so many things at once that the reader (or at least, both of us) gets lost. One could easily*
*imagine any of the 5 aims to be a paper in and of itself. We found it at times very hard to follow*
*even with our (relatively) similar backgrounds and interests, and in its current form it seems*
*inaccessible to most readers. There are multiple atlases, a new model, multiple lines of thought,*
*a somewhat surprising RSA analysis, all fleshed out to varying degrees. The amount of work is*
*impressive, yet it does not succeed in turning it into a coherent, followable paper.*

**[Response 1.10]** We are grateful to the reviewers for helping us improve our work's clarity and
accessibility. Thanks to this feedback we believe that we have substantially improved the quality
of our paper's writing. We have made the following changes:

a. Rewritten parts of the introduction, especially clarifying perspectives on ergodicity, and
providing a better intuition for the analysis to follow.

b. Reframed the aims to improve clarity.

c. Restructured the results, to make the paper more readable: The results now start with reaction
time based analysis that does not incorporate PRAD results, then moves to the corresponding
PRAD related results, providing a natural expansion from model-free to model-based analysis.
Stability and robustness have been moved to the end of the results section.

413 d. Expanded the analysis to include more detailed robustness analysis. Please see Results lines
414-451, and SI S6, S7, S8.

e. Shifted the functional hierarchy of nonergodicity section to the SI.

f. Rewritten parts of the discussion.

We have used the multiple atlases only to provide another layer of robustness analysis. We think
that this additional robustness check meaningfully improves the trustworthiness of our findings.

Overall, we have reduced our focus to 3 primary aims, with the fourth aim being the replicability
of our findings from the first 3 aims, and each aim has been treated in greater detail. We hope
that our paper now provides a much clearer and more followable flow of thought.

[Comment 1.11] *Minor points*

This paper does not have an excessive number of acronyms, yet this needs to be considered in the model preprint paper. For a paper so dependent on modeling the code should be made available at this stage, not upon publication.

[Response 1.11] We have now made the code available.

[Comment 1.12] *Figure 3 (and all neuroimaging figures) are using the Jet colormap. This is 1) non-perceptually uniform, 2) not grayscale friendly, or 3) colorblind friendly. See Crameri et al., 2020 Nat. Comm. on the issues regarding these color maps.*

[Response 1.12] Thank you for noticing this shortcoming in our visualizations. We have updated all of our colormaps to Crameri's own Vik colormap (<https://zenodo.org/records/8409685>).

[Comment 1.13] *On the whole, this is a paper on an important topic with an impressive amount of work, but that ultimately fails to convince through the sheer disparity of analyses which necessitate overly condensed, sometimes superficial treatments, and leave open many of the obvious challenges. It is possible that some of the issues we raise were in fact treated in some way - but we can assure the authors that both of us made a genuine effort to (re)read the paper closely.*

[Response 1.13] We thank the reviewers for their detailed and thoughtful feedback. We have tried to address their concerns by rewriting our paper. The revised paper has focused on a smaller set of points but expanded the level of analysis for these. Some of the additional analysis that did not contribute to the central theme of the paper has been moved to the SI.

[Comment 1.14] *The authors have not shared their code (they state they will do so upon publication, but I cannot see any compelling reason for this decision).*

[Response 1.14] We have now made the code available.

**Reviewer #2**

**[Comment 2.1]** *This is a very interesting study on ergodicity by Mistry et al, which examined*
*between-person (BP) and within-person (WP) brain-behavior associations using a*
*computational model of the stop signal task in ABCD data. Mistry et al. found robust evidence of*
*nonergodicity and Simpson’s Paradox, such that the BP and WP relationships had different*
*directions. The biggest strength of this study is its direct ability to examine WP processes. I will*
*focus my review mostly on ergodicity as I believe my expertise is more useful there (the*
*computational task modeling and fMRI analyses all appear excellent). The manuscript reads*
*well, the study is analytically impressive, and the results have important implications for*
*understanding of cognitive control processes at an individual-level. Moreover, the overall theme*
*relates to a growing emphasis on precision imaging and ergodicity in cognitive neuroscience.*

**[Response 2.1]** We thank the reviewer for their positive appraisal of our work.

**[Comment 2.2]** *However, there are fundamental misunderstandings of ergodicity throughout the*
*manuscript, perhaps related to not including key references (see comment #1). These*
*fundamental issues led to serious flaws in framing the issue of nonergodicity and how analyses*
*were performed. Major issues include not recognizing what non-ergodicity means (all-or-*
*nothing), leading to a focus on testing it rather than the importance of WP processes (comment*
*#2), the authors making their own fallacy in assuming ergodicity when averaging WP results*
*(comment #4), and conflating nonergodicity and Simpson’s Paradox in their analyses (comment*
*#9). There are additional major concerns related to framing and which analyses were chosen as*
*main vs. robustness, described below. While these are major issues, they are quite*
*understandable (and common mistakes) as ergodicity is a difficult concept to understand and is*
*relatively new to cognitive neuroscience. I believe they are all addressable issues and this could*
*be an important manuscript if addressed. I would look forward to reviewing a revised version of*
*this manuscript.*

**[Response 2.2]** We are grateful to the reviewer for their insights and have accordingly made
changes to the manuscript, as detailed below.

**[Comment 2.3]** *Major Comments*

*Introduction*

*1. Lines 55-57: “Nonergodicity [...], its application to human brain function remains*
*unexplored” is incorrect. While this is one of the few studies to empirically test nonergodicity in*
*this field, there are several articles that have explicitly focused on ergodicity in brain*
*functioning. Most of these articles were not cited in the manuscript despite being highly relevant.*
*Beyond it being responsible to cite these papers as they are of utmost relevance, referencing*
*them is also needed as key takeaways from these articles would aid some of the nuanced*
*misunderstandings in the present manuscript (see other points below). Relevant citations*
*include, chronologically:*

*a. Medaglia, J. D., Ramanathan, D. M., Venkatesan, U. M., & Hillary, F. G. (2011). The*
*challenge of non-ergodicity in network neuroscience. Network (Bristol, England), 22(1–4), 148–*
*153. <https://doi.org/10.3109/09638237.2011.639604>*

*b. Kraus, B., Zinbarg, R., Braga, R. M., Nusslock, R., Mittal, V. A., & Gratton, C. (2023).*
*Insights from personalized models of brain and behavior for identifying biomarkers in*
*psychiatry. Neuroscience & Biobehavioral Reviews, 152, 105259.*
*<https://doi.org/10.1016/j.neubiorev.2023.105259>*
*c. Hunter, M. D., Fisher, Z. F., & Geier, C. F. (2024). What ergodicity means for you.*
*Developmental cognitive neuroscience, 68, 101406. <https://doi.org/10.1016/j.dcn.2024.101406>*
*d. Gell, M., Noble, S., Laumann, T. O., Nelson, S. M., & Tervo-Clemmens, B. (2024). Psychiatric*
*neuroimaging designs for individualised, cohort, and population studies.*
*Neuropsychopharmacology, 1–8. <https://doi.org/10.1038/s41386-024-01918-y>*
*e. Mattoni, M., Fisher, A. J., Gates, K. M., Chein, J., & Olino, T. M. (2025). Group-to-individual*
*generalizability and individual-level inferences in cognitive neuroscience. Neuroscience &*
*Biobehavioral Reviews, 169, 106024. <https://doi.org/10.1016/j.neubiorev.2025.106024>*

**[Response 2.3]** We thank the reviewer for this comprehensive list of highly relevant citations,
and agree that our choice of wording was incorrect and have revised it:

“While progress has been made in understanding nonergodicity in behavior¹⁻⁶ and
establishing strong theoretical frameworks⁷⁻¹⁰, few studies have empirically addressed
nonergodicity in human brain function^{9,11,12}.” (Introduction lines 72-75)

Moreover, we have substantially revised the Introduction and Discussion to properly situate our
work within this existing literature. Please see multiple changes to Introduction lines 40-85.

**[Comment 2.4]** *2. A nuanced, but critical, misunderstanding (in my view) throughout the*
*introduction is the goal of testing nonergodicity. In my view, it is neither important nor*
*interesting to explicitly test for nonergodicity in human processes. Molenaar (2003) established,*
*both theoretically and through a statistical ‘proof’, that essentially every human process is*
*inherently nonergodic. Critically, ergodicity is all-or-nothing, requiring homogeneity between*
*persons and stationarity within persons. From Molenaar 2003 alone, we can confidently assume*
*that cognitive control processes would be nonergodic. Thus, if the goal is simply testing*
*nonergodicity, this manuscript does not provide much incremental knowledge. However, this is*
*really just an issue of framing. Testing the within-person process itself (rarely studied) is*
*interesting and important, and the results here are indeed a striking and useful demonstration*
*for the field to recognize that between-person processes are distinct and independent from*
*within-person processes (an issue of group-to-individual generalizability, slightly different from*
*ergodicity, see Fisher 2018). Unless the authors have a strong argument otherwise, I believe*
*much of the introduction needs reframing based on this nuance, perhaps more toward an*
*emphasis on why within-person study is important (and distinct from between-person study).*
*Several of the references in comment #1 could aid this reframing.*

**[Response 2.4]** We thank the reviewer for this crucial conceptual clarification, addressing which
has substantially improved our manuscript. We agree that the framing needed revision and that
the value of our work lies not in testing whether cognitive processes fail to meet the strict
definition of ergodicity (which, as the reviewer notes, can be theoretically assumed based on
Molenaar 2003), but rather in demonstrating what within-subject analyses reveal about cognitive
control mechanisms and how substantively these insights differ from between-subject findings.

Accordingly, in response to this comment, as well as related feedback from Reviewer 1, we have
extensively rewritten the Introduction and Discussion. These revisions shift the emphasis to: (1)
importance of within-person analysis for understanding cognitive mechanisms; (2) substantive
insights revealed by within-person brain-behavior associations; (3) degree to which within-
person and between-person inferences diverge; and (4) complementary nature of these two levels
of analysis for answering different scientific questions – all within the conceptual framing of
nonergodicity. We are grateful to the reviewer for pushing us toward this more precise and
accurate framing, which substantially strengthens the manuscript.

**[Comment 2.5]** 3. Lines 112-118: “[...] this conventional approach implicitly assumes
ergodicity [...].” While there are relevant ergodic assumptions (which the authors overlook
themselves in the present analyses, see comment 4), the intended point here is not necessarily
true. The assumption of ergodicity would only be relevant if the between-person study was used
for within-person conclusions (again, the within-person aspect is the key here and not
sufficiently addressed in the introduction). If researchers use between-person SST tests to make
between-person conclusions, nonergodicity is not highly relevant as the authors describe here.

**[Response 2.5]** We thank the reviewer for this important clarification. We have now revised the
manuscript to clarify that ergodicity assumptions are only problematic when between-person
findings are used (implicitly or explicitly) to draw within-subject conclusions and to emphasize
the implicit generalization that commonly occurs in cognitive neuroscience research. For
example:

“Conventionally, studies investigating the neural basis of inhibitory control have relied
on between-subjects analyses of SSRT, correlating individual differences in SSRT with
individual differences in task-related brain activation¹⁸⁻²⁵. These between-subjects
associations are often implicitly interpreted as reflecting within-subjects mechanisms.
However, this implicit generalization from between-subjects to within-subjects levels
critically depends on the assumption of ergodicity^{4,5,7,26}.” (Introduction lines 114-119)

**[Comment 2.6]** Main

4. The authors make their own ergodic assumption fallacy in their tests of within-person
processes. From my understanding, they estimated within-person relationships entirely
ideographically and then aggregated them into a single within-person effect size. Importantly,
ergodicity is not the equivalence of the average within-person relationship with the between-
person relationship, but the equivalence of literally every single person’s within-person
relationship with the between-person relationship (hence, why we can safely assume human
processes are non-ergodic without explicitly testing them). This point is made clear in Molenaar
2003, Fisher 2018, Hunter 2024, and Mattoni 2025, particularly with the emphasis of the
homogeneity assumption for ergodicity. At the minimum, it would likely be more appropriate to
instead present a distribution of the individual’s within-person estimates. If the authors are truly
focused on a single aggregated value, alternative approaches such as the two-stage random
effects meta-analysis presented by Lee and Gates 2024
(<https://doi.org/10.1080/00273171.2023.2229310>) would be more appropriate than the
approach used here.

 **Supplementary Figure S2. Distributions over subjects of within-subjects associations. a-d,**
 Distributions across subjects are shown for the within-subjects associations with brain activity of
 behavioral measures: Go RT (a), SSRT (b), probability of proactivity (c), and proactive delaying
 (d). Dashed lines in the half-violin plots indicate the 25th, 50th, and 75th percentiles, and dots
 denote individual subjects.

[Response 2.6] We thank the reviewer for this important methodological point, which has led us
to substantially strengthen our presentation of within-subjects results.

First, we now present the distributions of within-subjects estimates (SI Figure S2).

Second, we performed the 2 stage random effects meta-analysis of Lee and Gates:

“To summarize [within-subjects] distributions, we have reported their simple Cohen’s
*d*’s throughout this paper. However, this simple method of aggregating within-subjects
results does not account for heterogeneity across subjects of the variance of within-
subjects results. So, to test the robustness of our simple aggregated within-subjects
results, we compared them to aggregated results from a 2 stage random effects meta-
analysis that considers heterogeneity in the variance of subject-level estimates²⁷. We
found very high similarity between these 2 methods of aggregating within-subjects
results (SI Figure S8). The Pearson correlation between simple averages or Cohen’s *d*’s
versus the corresponding aggregated results using the 2 stage random effects meta-
analysis ranged from 0.987 to 0.998.” (SI Supplementary Results lines 81-90)

**Supplementary Figure S8. Aggregation of within-subjects associations was robust to use of**

**2-stage random effects meta-analysis. a-b,** Aggregated results from 2-stage random effects

meta-analysis were compared to simple aggregated results: average (a) and Cohen’s *d* (b). The

results that were aggregated were the distributions across subjects of the within-subjects brain-

behavior associations for Go RT, SSRT, probability of proactivity, and proactive delaying. The

2-stage random effects meta-analysis accounted for between-subjects heterogeneity in the

precision of within-subjects estimation. The simple average was the mean and the simple

Cohen’s *d* was the mean divided by the standard deviation. Dots denote voxels. Go RT, SSRT,

and probability of proactivity $N = 4423$; proactive delaying $N = 4137$.

Since the similarity was so high between our original approach to aggregating within-subjects
results and this more complex, model-based alternative, we have kept the original, simpler
approach for presenting our findings.

We agree that “ergodicity is not the equivalence of the average within-person relationship with
the between-person relation, but the equivalence of literally every single person’s within-person
relationship with the between-person relationship.” Still, we find it useful to present a single,
aggregated value for within-person relationships. When the average between- and within-person
relations *agree*, this does not imply *ergodicity*, however, when the average between- and within-
person relations *disagree*, this does imply *nonergodicity*. Thus, given the frequent disagreement
of average between- and within-person relations, comparing these both provides a striking
demonstration of nonergodicity and highlights the need for within-person analysis in addressing
within-person questions.

**[Comment 2.7]** 5. *The authors were focused on the sign of the relationship (i.e., positive vs.*
*negative) throughout the manuscript. Another striking finding is that the within-person effect*
*sizes were substantially higher. Can the authors speculate as to why this may be, and potentially*
*address this in the manuscript? Importantly, ergodicity is not simply the same direction of the*
*relationship, but literal equivalence.*

**[Response 2.7]** We focused on directional reversals (sign differences) because these represent
the most striking and scientifically consequential form of nonergodicity. When brain-behavior
relationships reverse direction across levels of analysis—exemplifying Simpson's paradox—this
fundamentally alters our mechanistic interpretation. For example, between subjects, proactive
delaying was positively correlated with activation in the anterior salience network (individuals
who used more proactive delaying had more activation in this network). In contrast, within
subjects, proactive delaying was negatively correlated with activation in the anterior salience
network (when individuals used more proactive delaying, they had less activation in this
network).

We agree that our within-subjects effects were somewhat larger than our between-subjects
effects, though we are not sure why. One speculation is that there exist different general coupling
strengths between stable traits versus dynamic states. Between-subjects effects reflect stable trait
differences that must be inferred from aggregated data mixing heterogeneous cognitive states,
possibly attenuating associations. Within-subjects effects capture state fluctuations—moment-to-
moment changes in attention, strategic adjustments, and fatigue—that may drive stronger brain-
behavior coupling. A second speculation is that since psychological laws are primarily expressed
at the individual level⁶, individual-level relationships that are “closer to” those laws may tend to
be stronger than group-level relationships that have a more distal connection to those laws.
Overall, we are not aware of literature that has explored the causes of this intriguing difference
and it may make for interesting future work.

**[Comment 2.8]** 6. *Figure 5 would be improved by marking the point at which sample size =*
*number of trials (I could not find the number of trials in manuscript). As Fisher 2018 pointed*

*out, there are important power implications for differing observations for between-person (n)*
*and within-person (t) study. This should be explicitly addressed in the manuscript.*

**[Response 2.8]** We have now added the number of trials (T = 360 across 2 runs of 180 trials
each). We have made changes to the figure as suggested for Figure 8 (formerly Figure 5).

Moreover, we thank the reviewer for highlighting the importance of the number of time points in
within-subjects study, and have now addressed the power implications of this parameter
explicitly in the manuscript:

“We note that while our study’s large number of subjects is important for assessing the
stability and generalizability of our inferred within-subjects relationships, the power to
detect a particular subject’s relationship depends not on the number of subjects but on the
number of temporal observations per subject (in our case, the number of trials)¹. Thus,
within-subjects research benefits from datasets that are large-scale in the number of
subjects as well as large-scale in the number of time points collected.” (SI Supplementary
Results lines 117-122)

**[Comment 2.9]** *7. The authors provide a useful robustness check in the supplement focusing*
*between-person SSRT study on stop trials and proactivity stud on all trials, which they state*
*“aligns more closely with our within-subjects analyses.” While the results are mostly consistent*
*with the main results presented, they are not as striking and it is unclear why this approach is a*
*robustness check instead of the main analysis, when the goal of the present paper is to compare*
*BP and WP relationships.*

**[Response 2.9]** We have selected this as the main analysis because our focus is on addressing the
scientific questions typically raised in the study of inhibitory control—hence the comparison is
based on the practical set of analysis that researchers typically conduct in between-subjects
analysis of this task^{18,21-25}, rather than a hypothetical alternate analysis that is not typically
conducted. We would like to highlight that this alternate comparison also shows strong
differences in BP and WP relationships.

**[Comment 2.10]** *8. Relatedly, supplementary Figure S2 panel C, this between-person result*
*seems to much more closely resemble the within-person results in Figure 2. This is not consistent*
*with the author’s description of robustness checks in the main manuscript. This needs explicit*
*clarification.*

**[Response 2.10]** We agree that comparing the between-person map formerly in S2c (now S5c)
and the corresponding within-person map formerly in S2d (now S5d) seems to give more
similarity than some of the other comparisons. However, we think that these maps still disagree
substantively, with for example opposite associations in the anterior cingulate cortex, left and
right insula, and precuneus.

**[Comment 2.11]** *9. Aim 5, “Brain networks exhibit varying degrees of nonergodicity and a*
*hierarchical organization by nonergodicity” is flawed; the authors erroneously conflate*
*Simpson’s Paradox and nonergodicity. Again, nonergodicity is all-or-nothing and ergodic*

*assumptions are violated with ANY non-equivalence, not necessarily a difference in relationship*
*direction (Simpson's Paradox). If the author's maintain the present analysis, this should be*
*renamed to measuring Simpson's paradox. That said, I'm not sure why quantifying where*
*Simpson's paradox occurs more often is informative. Instead, it may be more useful to*
*understand where in the brain WP relationships are +, WP relationships are -, BP relationships*
*are +, BP relationships are -. The point of ergodicity is that BP and WP relationships should be*
*considered separately.*

**[Response 2.11]** Following this, as well as comments from reviewer 1, this analysis has been
removed from the main results to the SI. We thank the reviewer for pointing out this imprecision
in our wording and we have now changed our wording appropriately. In other figures (Figures 3,
5, and 7), we highlight where WP and BP relationships are positive and negative.

**[Comment 2.12]** *Discussion*

*10. Much of the discussion would need reframing based on the context I described in comment*
*#2.*

**[Response 2.12]** The discussion has been updated based on the above comments to clarify that
our goal is to understand whether violations of ergodicity are substantive, meaning for example
that assuming ergodicity and generalizing between- to within-subjects analysis would not merely
change inferences by degree but would result in completely different inferences. Please see for
example:

"The central question we addressed is whether empirical brain-behavior relationships
underlying cognitive control are nonergodic, and critically, whether any nonergodicity
substantively challenges our understanding of inhibitory control mechanisms based on
traditional between-subjects analysis." (Discussion lines 457-460)

We have also updated the discussion to emphasize that our results illustrate that between-
subjects processes are distinct and independent from within-subjects processes. Please see for
example:

"Our findings challenge the implicit assumption of ergodicity in cognitive neuroscience
and demonstrate that examining how brain activity and behavior co-vary within
individuals over time provides fundamentally different insights than examining how they
co-vary across individuals." (Discussion lines 472-475)

**[Comment 2.13]** *11. Overall, the strength of this study is its unique ability to examine WP brain-*
*behavior relationships. More time should be spent explaining why that is important, what the*
*authors learned that BP study cannot tell us, and implications of those WP relationships.*

**[Response 2.13]** We have added this perspective to the discussion. Please see changes to
Discussion lines 556-629.

**[Comment 2.14]** *12. Related to the importance of WP study, perhaps the authors could*
*speculate on important directions forward for that goal, whether in terms of study design or*
*computational approaches.*

**[Response 2.14]** Please see changes to Discussion lines 673-675.

**[Comment 2.15]** *Minor*
*1. Figure 1 panels A-C are very useful. More attention could perhaps be spent on Panel D,*
*which did not provide me much clarity on the study goals without several looks and reads of the*
*caption.*

**[Response 2.15]** We have redesigned Figure 1D and hope that its new structure presents the
study's aims more clearly.

**[Comment 2.16]** *2. Lines 67-70: "Group-level relationships may obscure the dynamic processes*
*occurring within each individual [...]." More important nuance here. It is not so much that*
*group-level relationships obscure individual-level relationships, but that they are entirely*
*distinct and the study of between-person relationships cannot be used to inform individual-level*
*relationships without ergodic assumptions.*

**[Response 2.16]** We agree and have changed our wording to better reflect this nuance. For
example:

"Many important questions in the study of the brain and behavior are posed at the within-
individuals level⁶, and answering these questions requires either within-individuals
analyses, or the strong assumption of ergodicity^{1,4}." (Introduction lines 56-59)

**[Comment 2.17]** *3. Lines 74-76: "[...] paving a way for personalized interventions [...] there is*
*a pressing need to integrate [...] in inhibitory control." While I agree with the notion, this*
*statement was not well set up by the introduction, which spent little time on why nonergodicity is*
*important for the study of within-person processes, which I believe is what the authors are*
*referencing here.*

**[Response 2.17]** We agree and have changed significant parts of the introduction to reflect this.
For example:

"The implications of nonergodicity in neural and behavioral science are profound.
Analyzing data from a single individual over time can yield insights that cannot be
obtained by analyzing data across multiple individuals at a single point in time^{1,4,28}."
(Introduction lines 68-70)

**[Comment 2.18]** *4. I find the section "Adaptive regulation of inhibitory control associated with*
*distinct within-subjects results over subgroups" very interesting; it relates individual differences*
*in the WP brain-behavior processes (rather than averaging them) to real behavioral differences.*
*This was a joy to see and may be worth emphasizing more in the manuscript.*

**[Response 2.18]** Thank you, we have emphasized this more in the manuscript, for example in the
Discussion subsection “Individual differences in adaptive regulation of inhibitory control shape
individuals’ brain-behavior associations” (Discussion lines 590-629).

**[Comment 2.19]** *5. I’m not sure what “a paradigm shift [...] toward nonergodicity
neuroscience” (line 671) means. That read a bit grandiose.*

**[Response 2.19]** We have changed our wording (Discussion lines 471-472).

**[Comment 2.20]** *The authors provided example code for the computational model they used for
the SST. This code appropriately describes their modeling. However, it is not specific to the
ABCD data and there is no code for the fMRI analysis. Since ABCD is public data, this code
would be a useful resource for the field if the authors are willing to provide it. As currently
provided, the code cannot be used to reproduce the results.*

**[Response 2.20]** We have now made all of the code available, both for interfacing the PRAD
model with the ABCD data and for the fMRI analysis. The computational model code can be
found at https://github.com/scsnl/2024_Mistry_PRAD/tree/main/PRAD_Behavioral_Dynamics
and all other code used in the study can be found at
https://github.com/scsnl/2025_nonergodicity_mistry_branigan.

**NOTE TO REVIEWERS**

During the revision, we noticed a mistake in the code that produced our subject list. There was
an error in the computation of framewise displacement, which was used to exclude subjects,
resulting in subjects being included in our analyses who should have been excluded and vice
versa. The mistake was corrected, a corrected subject list was created, and all results have been
updated using the corrected subject list. Our previous subject list included 4469 subjects; our
new subject list includes 4423 subjects; and 4193 of the subjects in the new list were also in the
old list (i.e., ~95% of our subject list remains the same). Our group-level results are only very
slightly changed, and none of our paper's conclusions is changed. We apologize for this mistake.

**References**

- Fisher, A. J., Medaglia, J. D. & Jeronimus, B. F. Lack of group-to-individual generalizability
is a threat to human subjects research. *Proc Natl Acad Sci U S A* **115**, E6106-E6115,
doi:10.1073/pnas.1711978115 (2018).
- Castro-Schilo, L. & Ferrer, E. Comparison of nomothetic versus idiographic-oriented
methods for making predictions about distal outcomes from time series data.
*Multivariate Behavioral Research* **48**, 175-207 (2013).
- Hamaker, E., Ceulemans, E., Grasman, R. & Tuerlinckx, F. Modeling affect dynamics:
State of the art and future challenges. *Emotion Review* **7**, 316-322 (2015).
- Molenaar, P. C. On the implications of the classical ergodic theorems: Analysis of
developmental processes has to focus on intra-individual variation. *Developmental*
*Psychobiology: The Journal of the International Society for Developmental Psychobiology*
**50**, 60-69 (2008).
- Molenaar, P. C. & Campbell, C. G. The new person-specific paradigm in psychology.
*Current directions in psychological science* **18**, 112-117 (2009).
- Curran, P. J. & Bauer, D. J. The disaggregation of within-person and between-person
effects in longitudinal models of change. *Annual review of psychology* **62**, 583-619
(2011).
- Medaglia, J. D., Ramanathan, D. M., Venkatesan, U. M. & Hillary, F. G. The challenge of
non-ergodicity in network neuroscience. *Network: Computation in Neural Systems* **22**,
148-153 (2011).
- Gell, M., Noble, S., Laumann, T. O., Nelson, S. M. & Tervo-Clemmens, B. Psychiatric
neuroimaging designs for individualised, cohort, and population studies.
*Neuropsychopharmacology* **50**, 29-36 (2025).
- Hunter, M. D., Fisher, Z. F. & Geier, C. F. What ergodicity means for you. *Developmental*
*Cognitive Neuroscience* **68**, 101406 (2024).
- Mattoni, M., Fisher, A. J., Gates, K. M., Chein, J. & Olino, T. M. Group-to-individual
generalizability and individual-level inferences in cognitive neuroscience. *Neuroscience*
*& Biobehavioral Reviews*, 106024 (2025).
- Kraus, B. *et al.* Insights from personalized models of brain and behavior for identifying
biomarkers in psychiatry. *Neuroscience & Biobehavioral Reviews* **152**, 105259 (2023).
- Kievit, R. A., Scholte, H. S., Waldorp, L. J. & Borsboom, D. Inter- and intra-individual
differences in fluid reasoning show distinct cortical responses. *bioRxiv*, 039412 (2016).
- Molenaar, P. C. A manifesto on psychology as idiographic science: Bringing the person
back into scientific psychology, this time forever. *Measurement* **2**, 201-218 (2004).
- Mistry, P. K., Warren, S. L., Branigan, N. K., Cai, W. & Menon, V. Computational
Modeling of Proactive, Reactive, and Attentional Dynamics in Cognitive Control. *bioRxiv*,
2024.2010.2001.615613 (2024).
- Button, K. S. *et al.* Power failure: why small sample size undermines the reliability of
neuroscience. *Nat Rev Neurosci* **14**, 365-376, doi:10.1038/nrn3475 (2013).
- Marek, S. *et al.* Reproducible brain-wide association studies require thousands of
individuals. *Nature* **603**, 654-660 (2022).

Cai, W. *et al.* Both reactive and proactive control are deficient in children with ADHD
and predictive of clinical symptoms. *Translational psychiatry* **13**, 179 (2023).

Cai, W. *et al.* Hyperdirect insula-basal-ganglia pathway and adult-like maturity of global
brain responses predict inhibitory control in children. *Nature communications* **10**, 4798
(2019).

Zhang, S. *et al.* Independent component analysis of functional networks for response
inhibition: Inter-subject variation in stop signal reaction time. *Human brain mapping* **36**,
3289-3302 (2015).

Boehler, C. N., Appelbaum, L. G., Krebs, R. M., Hopf, J.-M. & Woldorff, M. G. The
influence of different Stop-signal response time estimation procedures on behavior–
behavior and brain–behavior correlations. *Behavioural brain research* **229**, 123-130
(2012).

Li, C. S., Huang, C., Constable, R. T. & Sinha, R. Imaging response inhibition in a stop-
signal task: neural correlates independent of signal monitoring and post-response
processing. *J Neurosci* **26**, 186-192, doi:10.1523/JNEUROSCI.3741-05.2006 (2006).

Li, C. S. *et al.* Neural correlates of impulse control during stop signal inhibition in
cocaine-dependent men. *Neuropsychopharmacology* **33**, 1798-1806,
doi:10.1038/sj.npp.1301568 (2008).

Chevrier, A. & Schachar, R. J. BOLD differences normally attributed to inhibitory control
predict symptoms, not task-directed inhibitory control in ADHD. *J Neurodev Disord* **12**, 8,
doi:10.1186/s11689-020-09311-8 (2020).

Chaarani, B. *et al.* Baseline brain function in the preadolescents of the ABCD Study. *Nat*
*Neurosci* **24**, 1176-1186, doi:10.1038/s41593-021-00867-9 (2021).

Aron, A. R. & Poldrack, R. A. Cortical and subcortical contributions to Stop signal
response inhibition: role of the subthalamic nucleus. *J Neurosci* **26**, 2424-2433,
doi:10.1523/jneurosci.4682-05.2006 (2006).

Mangalam, M. & Kelty-Stephen, D. G. Point estimates, Simpson's paradox, and
nonergodicity in biological sciences. *Neurosci Biobehav Rev* **125**, 98-107,
doi:10.1016/j.neubiorev.2021.02.017 (2021).

Lee, S. A. & Gates, K. M. From the individual to the group: Using idiographic analyses
and two-stage random effects meta-analysis to obtain population level inferences for
within-person processes. *Multivariate Behavioral Research* **59**, 1220-1239 (2024).

Molenaar, P. & Newell, K. M. *Individual pathways of change: Statistical models for*
*analyzing learning and development.* (American Psychological Association, 2010).
